# $V_0$: A Generalist Value Model for Any Policy at State Zero

**Yi-Kai Zhang** [1 2 3]   **Zhiyuan Yao** [4 3]   **Hongyan Hao** [3]   **Yueqing Sun** [3]   **Qi Gu** [3]   **Hui Su** [3]   **Xunliang Cai** [3]
**De-Chuan Zhan** [1 2]   **Han-Jia Ye** [1 2]

## Abstract

Policy gradient methods rely on a baseline to measure the relative advantage of an action. In Actor-Critic LLM training (*e.g.*, PPO), this baseline is typically estimated by a Value Model as large as the policy. However, as the policy evolves, the value model requires expensive, synchronous training. To avoid this overhead, GRPO eliminates the value model by using the average reward of group rollouts as the baseline; yet, this necessitates extensive sampling to maintain stability. In this paper, we propose $V_0$, a Generalist Value Model capable of estimating the expected performance of any model on unseen prompts without parameter updates. We reframe value estimation by treating the policy's dynamic capability as an explicit context input, leveraging a history of instruction-performance pairs to dynamically profile the model. Acting as a resource scheduler, $V_0$ predicts success rates prior to rollout during GRPO training for efficient sampling budget allocation. During deployment, it functions as a router dispatching instructions to the most cost-effective model. Empirical results demonstrate that $V_0$ significantly outperforms heuristic budget allocation and achieves a Pareto-optimal performance-cost trade-off in LLM routing.

## 1. Introduction

In the post-training phase of Large Language Models (LLMs), Reinforcement Learning with Verifiable Rewards (RLVR) has emerged as a dominant paradigm (Yang et al., 2025a; DeepSeek-AI et al., 2025). A fundamental requirement of these policy gradient methods is a robust baseline

[1]School of Artificial Intelligence, Nanjing University [2]National Key Laboratory for Novel Software Technology, Nanjing University [3]Meituan, China [4]Zhejiang University. Correspondence to: Qi Gu <guqi03@meituan.com>, Han-Jia Ye <yehj@lamda.nju.edu.cn>.

*Proceedings of the 43rd International Conference on Machine Learning*, Seoul, South Korea. PMLR 306, 2026. Copyright 2026 by the author(s).

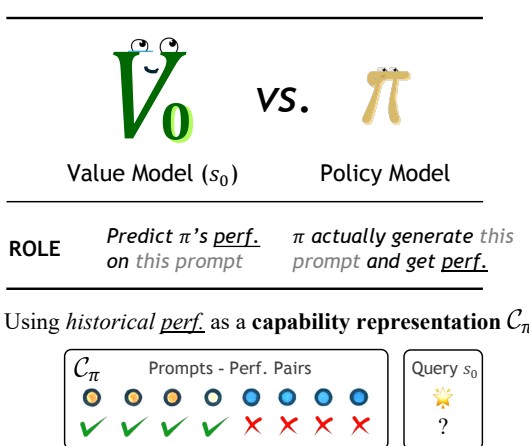

*Figure 1.* **The $V_0$ Paradigm**: Reframing Value Estimation as In-Context Prediction via Historical Capability Representation.

to estimate the relative advantage of an action, ensuring the model reinforces behaviors that outperform its current average capability. Traditionally, Actor-Critic architectures (*e.g.*, PPO) address this by maintaining a parameterized Value Model (Critic). While effective at variance reduction, this approach introduces a severe *coupling dilemma*: as the policy $\pi$ evolves, the value model $V^\pi$ must be synchronously and incrementally trained to track the non-stationary target, incurring massive computational costs and memory overhead (Yue et al., 2025; Liu et al., 2024). Group Relative Policy Optimization (GRPO) (Shao et al., 2024) eliminates the independent value model entirely, instead approximating the baseline via the mean reward of group rollouts. However, this essentially shifts the cost from training to sampling: to prevent high variance or reward collapse (where rewards become uniform zeros or ones) in complex tasks, GRPO necessitates extensive Monte Carlo sampling, creating a new bottleneck in computational efficiency and training stability (Li et al., 2025b; Zheng et al., 2025a; Fu et al., 2025).

In this paper, we propose $V_0$, a generalist value model designed to resolve this efficiency-stability trade-off. Our core

insight is to reframe value estimation: instead of treating the policy $\pi$ as an implicit variable hidden within the value model's parameters (requiring continuous training), we treat it as an explicit context input $\mathcal{C}_\pi$. Formally, this shifts the estimation paradigm from a parameterized function $V^\pi(s_0)$ to a conditional prediction $V(\mathcal{C}_\pi, s_0)$, where $\mathcal{C}_\pi$ consists of a sequence of historical query-performance pairs. This design allows $V_0$ to dynamically *read* the current capabilities of any policy without gradient updates, effectively decoupling value estimation from policy evolution. In this paper, we focus specifically on State Zero (*i.e.*, the initial prompt), and $V_0$ acts as a strategic resource scheduler: in the context of GRPO training, it predicts success probabilities prior to roll-out, enabling adaptive budget allocation to avoid wasteful sampling on effectively solved or unsolvable tasks; in model deployment, aligning with inference-time compute scaling, it serves as a router to dispatch instructions to the most cost-effective model ensuring Pareto-optimal performance.

Implementing $V_0$ requires an architecture that can simultaneously comprehend semantic instructions and accurately infer statistical patterns from historical performance. Since standard LLMs often struggle with precise numerical estimation, we design a hybrid *Semantic-Perception to Structured-Reasoning* architecture. We employ an Embedding Backbone to map the capability history (as context) and the target query into high-dimensional semantic vectors. To bridge the gap between the high-dimensional, continuous nature of these embeddings with the requirement for precise, structured logical inference, we introduce a trainable *Residual Query Adapter*. This module extracts features correlated with model capability, projecting them into compact vectors for our TabPFN (Grinsztajn et al., 2025) inference head. Leveraging its pre-trained Bayesian inference capabilities, TabPFN treats the historical performance of $\pi$ as a reference set, performing in-context learning in a *single forward pass* to construct decision boundaries and infer success probabilities. Thus, $V_0$ moves beyond memorizing parameters to learning the *meta-knowledge* of capability estimation.

Achieving robust generalization across arbitrary policies, however, presents a fundamental challenge. Through a mutual information analysis of joint training, we discover that inherent capability gaps between policies cause the mutual information $I(Y; X, \mathcal{C})$ ($Y$ is performance, $X$ is query, and $\mathcal{C}$ is context), to decompose into a dominant "shortcut term" $I(Y; \mathcal{C})$. This implies that the model degenerates into a heuristic that judges policy strength based solely on the capability context, neglecting the critical interaction $I(Y; X \mid \mathcal{C})$. To mitigate this, we propose a composite loss strategy: a Pairwise Ranking Loss (based on the Bradley-Terry model) to enforce relative score separation within the same context, combined with soft cross-entropy to calibrate absolute probabilities. Empirical results demonstrate that $V_0$ exhibits reliable scaling potential and generalization. Our

main contributions are:

- We propose $V_0$, a framework that decouples the value model from policy parameters by reframing estimation as a conditional prediction problem, $V(\mathcal{C}_\pi, s_0)$. To realize this, we introduce a hybrid *Semantic-Perception to Structured-Reasoning* architecture, making such context-aware policy assessment feasible for the first time.
- We identify the shortcut bias in training $V_0$ via mutual information analysis and propose a composite objective (pairwise ranking loss + soft CE) to resolve it.
- We demonstrate that $V_0$ tracks the evolution during GRPO training, offering superior stability over coupled value models. Additionally, $V_0$ efficiently solves the cold-start problem in *Budget Allocation* and approaches the performance-cost Pareto frontier in *Inference Routing*.

## 2. Preliminaries

In this section, we formalize value estimation in LLM reinforcement learning as a conditional prediction task governed by in-context capability. We further introduce TabPFN as our inference backbone. For clarity, we denote the input query (state zero, $s_0$) as $x \in \mathcal{D}_{\text{prompt}}$.

**Value Estimation in Post-training RL.** We model this part as a Markov Decision Process (MDP) (Ramamurthy et al., 2023). Given a query $x$, a policy $\pi_\theta$ generates a response $y$, which is evaluated by a reward function $\mathcal{R}(x, y) \in \{0, 1\}$ in the context of RLVR. Traditional methods like PPO rely on a parameterized value function $V_\phi(x)$ to estimate the expected return. Distinct from Outcome Reward Models (ORMs) that approximate the ground-truth reward $\mathcal{R}(x, y)$ *after* generation (Shi et al., 2024; Li et al., 2025a), $V_\phi(x)$ aims to predict with the policy capability *before* generation. However, $V_\phi$ in PPO introduces a coupling dilemma: it must synchronously track the non-stationary distribution of the evolving $\pi_\theta$. While *value-free* methods like GRPO obviate this by approximating the baseline via group averages ($V(x) \approx \frac{1}{G} \sum R_i$), they suffer from high Monte Carlo variance, where sparse rewards often collapse into uniform values, yielding uninformative gradients.

**In-Context Capability Representation.** To resolve the coupling dilemma while retaining the variance-reduction benefits of value functions, we reframe value estimation from parameter fitting to *In-Context Learning (ICL)* (Hollmann et al., 2023). In this paradigm, the policy $\pi$ is no longer treated as a latent variable implicit in the weights $\phi$, but is explicitly represented as a context set $\mathcal{C}_\pi = \{(x_i, r_i)\}_{i=1}^N$ of historical query-performance pairs. Consequently, value estimation transforms into inferring the Posterior Predictive Distribution (PPD) (Müller et al., 2022)

for a target query $x$:

$$P(r \mid x, \mathcal{C}_\pi) = \int P(r \mid x, \mathcal{M}) P(\mathcal{M} \mid \mathcal{C}_\pi) \, d\mathcal{M} \quad (1)$$

where $\mathcal{M}$ represents the underlying capability model derived from observations. This shift allows $V_0$ to predict $V(x, \mathcal{C}_\pi)$ by perceiving the capability boundaries of any policy through its $\mathcal{C}_\pi$, enabling zero-gradient adaptation to unseen policies. This shares a similar idea with learnware (Zhou, 2016; Zhou & Tan, 2024; Tan et al., 2025).

## 3. Related Work

**Generalist Value Representation and In-Context Attempts.** To construct more robust value estimation, recent research has explored alternative parameterizations (Cohen et al., 2025; Yan et al., 2025), such as VRPO (Zhu et al., 2025) and RELC (Cao et al., 2024), which derive intrinsic rewards. However, these models may suffer from feedback saturation when the policy's capability exceeds the critic's evaluation ceiling. In the realm of decoupled In-Context Learning (ICL), GVL (Ma et al., 2025) uses cross-task progress to evaluate objective states, whereas DVPO (Huang et al., 2025) attempts to probe capability via sequence distributions. DVPO may lack discriminative power when different policies generate similar response prefixes (see Appendix E for more cases). In contrast, $V_0$ explicitly encodes model capability by ingesting large historical contexts, synthesizing semantic and statistical dimensions to avoid these pitfalls.

**Predictive Capability Boundaries for Resource Allocation.** Our work targets value estimation at state zero, effectively predicting capability boundaries to optimize resource allocation across the LLM lifecycle. During training, $V_0$ functions as an adaptive budget allocator; unlike methods such as Knapsack RL (Li et al., 2025b) that rely on lagged evaluations, $V_0$ provides real-time, zero-gradient adaptation to prevent wasted compute on mastered or hard samples (Zheng et al., 2025b; Zeng et al., 2025; Yang et al., 2025b; Sun et al., 2025; Yao et al., 2026). During inference, it advances beyond routing based on latent representations (Zhuang et al., 2025; Chen et al., 2024b; Ong et al., 2024; Zhang et al., 2025c) or reference anchors (Zhang et al., 2025b; Jitkrittum et al., 2025) to explicit capability-aware dispatching, enabling the selection of the most economical model from a candidate fleet for any given prompt.

For more discussion, please refer to Appendix I.

## 4. Method

In this section, we formalize Generalist Value Estimation as a conditional prediction task and detail the $V_0$ framework. We first describe the transition from implicit parameter fitting to explicit contextual inference. We then introduce the

hybrid *Semantic-Perception to Structured-Reasoning* architecture, specifically the Residual Query Adapter that bridges high-dimensional semantics with Bayesian inference. Finally, we analyze the shortcut learning phenomenon via Mutual Information (MI) and derive a debiased objective.

### 4.1. Value Estimation as Contextual Inference

> *"It is what you do that defines you."*
> — *Batman Begins*

Traditional value models $V^\pi(x)$ are intrinsically coupled with a specific policy $\pi$, necessitating synchronous updates whenever the policy parameters shift. Inspired by the philosophy that an entity's identity is defined by its observable actions, we propose that a policy's capability is best characterized by its historical behavior. We break the coupling dilemma by reframing value estimation as an **In-Context Learning (ICL)** problem. Instead of embedding policy information into latent weights, we represent the capability of an arbitrary policy $\pi$ via an explicit context set $\mathcal{C}_\pi$ consisting of historical query-performance pairs:

$$\mathcal{C}_\pi = \{(x_i, r_i) \mid x_i \in \mathcal{D}_{\text{context}}, r_i \in \{0, 1\}\}_{i=1}^N \quad (2)$$

where $r_i$ denotes the binary success on query $x_i$. The objective of $V_0$ is to learn a generalist meta-function that infers the performance on a target query $x_t$ conditioned on $\mathcal{C}_\pi$:

$$\hat{v} = V_0(x_t, \mathcal{C}_\pi) \approx P(r_t = 1 \mid x_t, \mathcal{C}_\pi) \quad (3)$$

By treating $\pi$ as a capability input, $V_0$ achieves *zero-gradient adaptation* to unseen policies, allowing it to dynamically perceive capability boundaries without parameter updates.

### 4.2. $V_0$ Model Architecture

The implementation of $V_0$ requires bridging the gap between high-dimensional natural language semantics and low-shot statistical inference. As illustrated in Figure 2, our architecture consists of three components.

**Semantic-Perception Backbone.** To map discrete instructions into a continuous manifold, we utilize a pre-trained embedding encoder $f_{\text{enc}}$. For both context queries $\{x_i\}$ and the target $x_t$, we extract the global semantic representation $\mathbf{h} = \text{Pool}(f_{\text{enc}}(x)) \in \mathbb{R}^{d_{\text{embed}}}$. This step captures deep semantic features, domain attributes, and latent difficulty information, providing a rich semantic foundation.

**Residual Query Adapter.** Directly feeding $\mathbf{h}$ into a statistical head is problematic due to a *feature gap*: TabPFN is designed for structured tabular data (where specific columns represent fixed meanings like "age" or "income"), whereas LLM embeddings are highly entangled. We design the *Residual Query Adapter* as a "Semantic Prism." Just as a

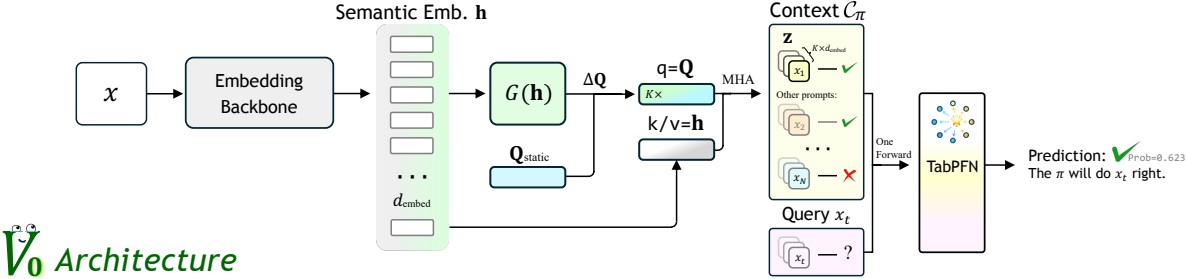

**$\overset{\smile}{V}_0$ Architecture**

*Figure 2.* **The $V_0$ Architecture.** A Semantic Backbone extracts embedding **h**, which the Residual Query Adapter projects into structured features using queries $\mathbf{Q}_{\text{static}}$ and dynamic $\Delta\mathbf{Q}$. After obtaining context $\mathcal{C}_\pi$ and query $x$, they are fed into the TabPFN inference head.

prism refracts natural light (entangled information) into distinct spectral bands, our adapter projects the mixed semantics of **h** into $K$ independent feature channels. We employ a set of learnable *Static Queries* $\mathbf{Q}_{\text{static}}$ to capture general capability dimensions (*e.g.*, "arithmetic complexity"). To handle instance-specific nuances, we introduce a residual mechanism where a generator $G$ produces dynamic offsets $\Delta\mathbf{Q}$ conditioned on **h** as $\mathbf{Q} = \mathbf{Q}_{\text{static}} + G(\mathbf{h})$. The final structured features **z** are obtained via Multi-Head Attention (MHA), using $\mathbf{Q}$ to probe the semantic backbone:

$$\mathbf{z} = \text{MHA}(\text{q=}\mathbf{Q}, \text{k/v=}\mathbf{h}) \in \mathbb{R}^{K \times d_{\text{embed}}} \quad (4)$$

This process ensures *column alignment*: the resulting **z** possesses a fixed coordinate system essential for Bayesian inference, where each dimension implicitly characterizes a consistent capability factor.

**Probabilistic In-Context Head.** We employ TabPFN as our inference core. TabPFN treats the transformed pairs $\{(\mathbf{z}_i, r_i)\}_{i=1}^N$ as observations. In a *single forward pass*, it approximates the posterior predictive distribution (PPD) (Müller et al., 2022) for the target:

$$\hat{r}_t \sim P(r \mid \mathbf{z}_t, \{(\mathbf{z}_i, r_i)\}_{i=1}^N) \quad (5)$$

Essentially, the head dynamically measures the statistical correlation between $\mathbf{z}_t$ and history $\{\mathbf{z}_i\}$ via attention, inferring the reward distribution without gradient updates.

### 4.3. A Mutual Information Perspective of Shortcuts

**Information Decomposition.** Although we enforce a global label balance $P(Y = 1) \approx 0.5$ to maximize entropy $H(Y)$, this does not imply conditional independence. The capabilities of different policies vary significantly, leading to a non-uniform conditional prior $P(Y = 1 \mid \mathcal{C})$. To analyze the optimization dynamics, we decompose the Mutual Information between the target label $Y$ and the joint input $(X, \mathcal{C})$:

$$I(Y; X, \mathcal{C}) = \underbrace{I(Y; \mathcal{C})}_{\text{Context Shortcut}} + \underbrace{I(Y; X \mid \mathcal{C})}_{\text{Causal Reasoning}} \quad (6)$$

Here, $I(Y; \mathcal{C})$ quantifies the information gain derived solely from the historical performance, independent of the current query $X$. Conversely, $I(Y; X \mid \mathcal{C})$ represents the robust reasoning capability that $V_0$ aims to learn. As formalized below in Theorem 4.1, minimizing cross-entropy loss naively encourages the model to exploit the context shortcut.

---

**Theorem 4.1.** *Let $\mu(\mathcal{C}) \triangleq P(Y = 1 \mid \mathcal{C})$ denote the latent capability prior of context $\mathcal{C}$. If $\text{Var}[\mu(\mathcal{C})] > 0$, then $I(Y; \mathcal{C}) > 0$, and we have:*

$$\min_\theta \mathcal{L}_{CE}(P(Y \mid \mathcal{C})) < H(Y).$$

*Thus, a model minimizing $\mathcal{L}_{CE}$ can strictly reduce error by fitting prior $\mu(\mathcal{C})$ alone, independent of the input $X$.*

---

**Debiasing via Shift-Invariant Ranking.** To decouple the prediction from the context prior $\mu(\mathcal{C})$, we introduce intra-context ranking. We define the output as a logit score $s(x, \mathcal{C})$, such that the final value probability is $V_0(x, \mathcal{C}) = \sigma(s(x, \mathcal{C}))$. We construct training pairs $(x_i, x_j)$ drawn from the *same* context $\mathcal{C}$ with opposing labels (where $y_i \succ y_j$), and minimize the Bradley-Terry ranking loss on the logits:

$$\mathcal{L}_{\text{rank}} = -\mathbb{E}_{\mathcal{C} \sim \mathcal{D}} \left[ \log \sigma \left( s(x_i, \mathcal{C}) - s(x_j, \mathcal{C}) \right) \right] \quad (7)$$

This objective forces the model to discriminate based on the relative difficulty of queries rather than the absolute capability of the policy.

---

**Theorem 4.2.** *Let $\tilde{s}(x, \mathcal{C}) = s(x, \mathcal{C}) + b(\mathcal{C})$ be a scoring function perturbed by an arbitrary context-dependent bias $b(\mathcal{C})$. The gradient of the ranking loss with respect to parameters $\phi$ of $V_0$ satisfies:*

$$\nabla_\phi \mathcal{L}_{rank}(\tilde{s}) = \nabla_\phi \mathcal{L}_{rank}(s).$$

---

Theorem 4.2 ensures that any shared context bias $b(\mathcal{C})$ is eliminated by the logit difference $s(x_i) - s(x_j)$. This invariance renders the ranking objective orthogonal to the subspace of context shortcuts $I(Y; \mathcal{C})$.

**Composite Optimization.**   While ranking eliminates bias, downstream tasks (e.g., risk-aware routing) require calibrated probabilities $V_0 \in [0, 1]$. We therefore optimize a hybrid objective to balance discrimination and calibration:

$$\mathcal{L} = \alpha\mathcal{L}_{\text{rank}}(s) + (1 - \alpha)\mathcal{L}_{\text{CE}}(V_0) \qquad (8)$$

This ensures $V_0$ learns the conditional interaction $I(Y; X \mid C)$ while maintaining probabilistic calibration. We provide a detailed analysis and proof of this section in Appendix A, followed by a diagnostic framework based on *residual orthogonality* in Appendix B for empirical verification.

## 5. Experiments

In this section, we evaluate the performance of $V_0$ as a Generalist Value Model. We focus on two pivotal questions: (**1**) Stability: Can $V_0$ maintain consistent estimation accuracy despite the distribution shifts inherent in RL training? (**2**) Generalization: Does $V_0$ achieve zero-shot generalization across unseen prompts, policy models of varying capabilities, and domains with diverse difficulties and semantics?

**Implementation Details.**   $V_0$ employs a frozen Qwen3-Embedding-0.6B as the semantic backbone ($d_{\text{embed}} = 1024$) and utilizes TabPFN-v2.5 as the inference head. The *Residual Query Adapter* is configured with 168 static queries, a projection dimension of 6, and 3 MHA heads. During our main experiments, we fine-tune the adapter while keeping the backbone and TabPFN frozen. We train using a batch size of 2. For each training and test instance, we sample a context size of $N = 256$ pairs from the context pool, and use a query batch size of 8. The objective balances the Pairwise Ranking Loss and Soft Cross-Entropy ($\alpha = 0.25$), optimized via AdamW with a learning rate of 2e-4.

**Policy Zoo and Data Construction.**   Unlike standard value models trained on static prompt, $V_0$ learns from *policy behaviors*. We conduct GRPO training on the DAPO-Math-17k (Yu et al., 2025) dataset using three distinct architectures: DeepSeek-R1-Distill-Qwen-1.5B (DeepSeek-AI et al., 2025), Qwen3-4B-Thinking (Yang et al., 2025a), and *Qwen2.5-7B-Instruct* (Yang et al., 2024a). We capture policy checkpoints every 1,024 samples along the training trajectory, resulting in approximately 247 checkpoints per architecture. For every checkpoint and query $x$, we estimate the ground truth value $r \in [0, 1]$ using avg@10 (the average success rate over 10 stochastic rollouts). We apply a threshold of 0.5 to demarcate positive and negative samples for classification metrics, while retaining continuous values for regression tasks. This process yields a comprehensive data pool where every step includes:

1. $\mathcal{D}_{\text{on-policy}}$: The actual on-policy rollouts generated during the GRPO training process.
2. $\mathcal{D}_{\text{held-out}}$: Over 10k additional rollouts performed on held-out queries for each checkpoint.

From above data pool, we design two protocols:

**Sequential Alignment (Simulating Standard RL).** This setting mimics a real-world RLVR run where $V_0$ must track a continuously evolving policy. We align the training of $V_0$ with the policy's training timeline.

1. **Test Set:** We reserve the fixed set $\mathcal{D}_{\text{on-policy}}$ at each training step exclusively for evaluation.
2. **Context Pool:** From the remaining $\mathcal{D}_{\text{held-out}}$, we split about half to the context pool. This pool retains the natural distribution of the policy (potentially imbalanced), reflecting the raw observation stream available during RL. We sample 256 pairs as context from this pool.
3. **Training Set:** From the remaining part of $n_{\text{extra}}$, we sample query-response pairs for training. As mentioned in the subsection 4.3, we balance positive and negative queries. The total number varies between 200 and 800 per step depending on the policy's capability.

Results reported in Table 1 and Figure 3 utilize this setting.

**Strict Generalization (Zero-Shot Transfer).**   To test if $V_0$ simply memorizes specific prompts, we enforce a strict separation based on query IDs across the entire timeline.

1. **Test Set:** We partition the set of unique query IDs into two disjoint sets. One is reserved strictly for testing to ensure that a test query is never encountered during training, even if different steps involve different labels. There are approximately 200 queries per step corresponding to the reserved Test IDs.
2. **Context Pool:** From the samples associated with the non-test IDs, we allocate about half to the context pool (providing $\approx 1500$ candidates per step). As in the above setting, we preserve the natural distribution.
3. **Training Set:** The remaining queries from non-test IDs are used for training, balanced 1:1 (positive/negative), yielding 200–800 samples per step.

Please refer to Appendix F for more details.

**Baselines.**   We benchmark $V_0$ against four baselines representing different paradigms of value estimation:

- **Vanilla Value Model (Coupled):** Represents the standard PPO value function. We append a linear head (from embedding dim $\rightarrow 1$) to the last token of the LLM with the same architecture as the policy. We perform a cold start (5 epochs on the base model rollouts) followed by incremental full-parameter fine-tuning (5 epochs per step) on the evolving trajectory. We use an MSE loss, batch size 32, and learning rate 1e-5.
- **Reward Model (Outcome-based):** We use *Qwen2.5-Math-RM-72B* (Yang et al., 2024b) to score the prompt directly. This baseline assesses the intrinsic difficulty of prompt rather than the specific capability of the policy.
- $k$**NN-Contextual:** A non-parametric baseline that main-

*Table 1.* **Performance Comparison of Value Estimation Methods** across Three Architectures during GRPO Training. We evaluate Intra-Group AUC (on the 1st epoch and all), Pairwise Accuracy, and Calibration MSE.

| Arch. & Metrics Method | DeepSeek-R1-Distill-Qwen-1.5B | | | | Qwen3-4B-Instruct-2507 | | | | Qwen2.5-7B-Instruct | | | |
|---|---|---|---|---|---|---|---|---|---|---|---|---|
| | Intra AUC | | Pair. Acc. | Calib. MSE | Intra AUC | | Pair. Acc. | Calib. MSE | Intra AUC | | Pair. Acc. | Calib. MSE |
| | 1st Ep. | All Eps. | | | 1st Ep. | All Eps. | | | 1st Ep. | All Eps. | | |
| Prev-Epoch | – | .835 | .419 | .445 | – | .860 | .427 | .331 | – | .728 | .374 | .776 |
| Reward Model | .540 | .539 | – | .624 | .613 | .629 | – | .594 | .659 | .693 | – | .613 |
| $k$NN-Contextual | .692 | .818 | .597 | .403 | .652 | .861 | .496 | .295 | .683 | .754 | .422 | .642 |
| Vanilla Value Model | .708 | .840 | .675 | .113 | .731 | .898 | .600 | .098 | .637 | .830 | .547 | .140 |
| Step-wise Retrain VM | .769 | .757 | .876 | .187 | .703 | .710 | .792 | .213 | .692 | .701 | .854 | .144 |
| $V_0$ (Ours) | **.887** | **.913** | **.940** | **.072** | **.893** | **.904** | **.884** | **.098** | **.883** | **.879** | **.956** | **.099** |

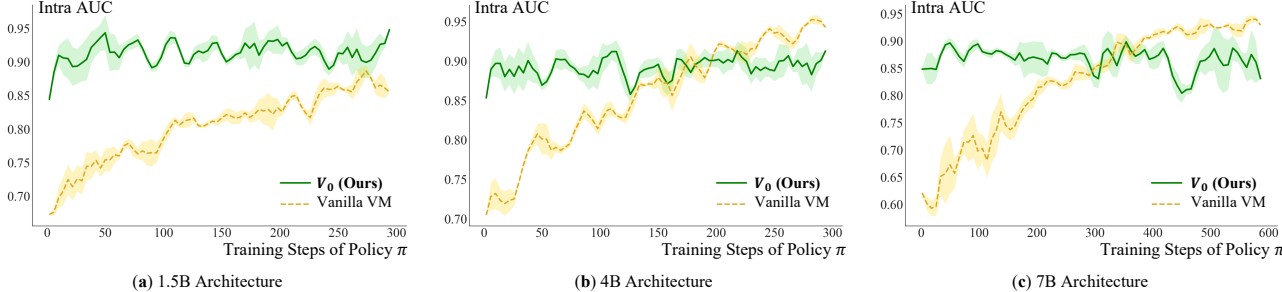

(a) 1.5B Architecture      (b) 4B Architecture      (c) 7B Architecture

*Figure 3.* **Comparison of Value Estimation Stability during Policy Training**. We track the estimation performance (Intra-AUC) of $V_0$ and the Vanilla VM across the training trajectories of three different architectures. The horizontal axis is the training steps of the policy model $\pi$. While the Vanilla VM exhibits a performance lag and instability, $V_0$ maintains high, consistent accuracy from the very first step.

tains a FIFO buffer (window size 2,048) of query-performance pairs. It estimates value by averaging on the $k = 64$ nearest neighbors with Euclidean distance.
- **Step-wise Retrain:** An reference where a fresh value model is trained *from scratch* at every single timestep using all available $\mathcal{D}_{\text{held-out}}$.

**Evaluation Metrics.** We employ three primary metrics to assess estimation quality: **(1) Intra-Context AUC:** Measures the model's ability to discriminate between successful and failed queries within the *same* policy checkpoint capability distribution. **(2) Pairwise Calibration Accuracy:** We construct pairs of the *same* query ID evaluated by *different* checkpoints ($\pi_i, \pi_j$). The model must correctly predict $P(r_i) > P(r_j)$ if and only if the ground truth satisfies $r_i > r_j$, testing the ability to track capability evolution. **(3) Calibration MSE:** The Mean Squared Error between the predicted probability and the ground truth `avg@10` reward. A detailed theoretical mapping between these metrics and MI components is provided in [subsection D.3](#).

**Applications in Resource Scheduling.** Beyond validating estimation accuracy, we evaluate the utility of $V_0$ in two critical resource scheduling scenarios: dynamic budget allocation during training and model routing during inference.

**Budget Allocation for Data Efficiency**

**Setting and Objective.** In Value-Free methods like GRPO, the baseline is estimated via group rollouts. Standard approaches typically assign a fixed budget of rollouts to every prompt. This strategy is suboptimal: for trivial prompts, the model yields rollouts that are entirely correct, while for distinctively hard ones, it yields rollouts that are entirely incorrect. In both extremes, the advantage collapses to zero, rendering these rollouts ineffective.

We formulate budget allocation as a constrained optimization problem maximizing the *Exploration Utility*:

$$\max_{\{B_i\}} \sum_i \text{Utility}(B_i, p_i) \quad \text{s.t.} \quad \sum_i B_i \leq B_{\text{total}} \quad (9)$$

where $B_i$ is the number of budget allocated to the $i$-th prompt, $p_i$ is the success rate, and $B_{\text{total}}$ is the global compute budget. The primary challenge is determining $p_i$. Existing heuristics rely on success rates from the previous epoch, but frequent policy updates make these historical metrics prone to latency. In contrast, $V_0$ predicts the current policy's success probability $p_i = P(r = 1 \mid x_i, \mathcal{C}_{\text{prev-step}})$ by utilizing the rollouts from the previous step as context. This provides an estimate based on the current capabilities. Crucially, this allows $V_0$ to allocate budget even for training samples with no prior history.

*Table 2.* **Strict Generalization Performance on Held-out Samples**. Unlike the standard setting, test samples here are excluded from the historical training trajectories of all previous steps to prevent memory overfitting.

| Method | DeepSeek-R1-Distill-Qwen-1.5B | | | Qwen3-4B-Instruct-2507 | | | Qwen2.5-7B-Instruct | | |
|---|---|---|---|---|---|---|---|---|---|
| | Intra AUC | Pair. Acc. | Calib. MSE | Intra AUC | Pair. Acc. | Calib. MSE | Intra AUC | Pair. Acc. | Calib. MSE |
| Vanilla Value Model | .560 | .467 | .267 | .512 | .304 | .474 | .527 | .507 | .583 |
| $V_0$ (Ours) | **.710** | **.895** | **.139** | **.689** | **.804** | **.138** | **.693** | **.840** | **.165** |

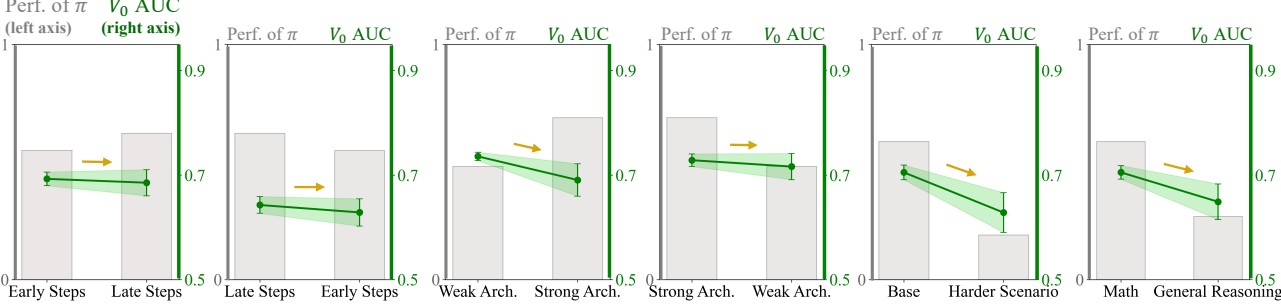

*Figure 4.* **Robust Generalization of $V_0$ Across Diverse Distribution Shifts**. The grey bars represent the performance of the policy $\pi$ (*left axis*), while the green lines denote the AUC of $V_0$ (*right axis*). $V_0$ is trained solely on the source distribution (*left*) and directly transferred to the unseen distribution (*right*). Despite fluctuations in policy training stages (Early Steps *vs.* Late Steps), model architectures (Weak Arch. *vs.* Strong Arch.), or task domains (Base *vs.* Harder/General), $V_0$ maintains a stable and high AUC.

**Utility Function and Solution.** Next, we define a utility function to quantify the training value of a prompt based on *Expected Gradient Signal Strength*. This is defined as the expected sum of absolute advantages within a single update step. This metric accounts for how the ratio of positive to negative prompts influences gradient. We derive the following closed-form approximation (please see Appendix C for the full derivation):

$$\text{Utility}(B_i, p_i) = B_i(1 - p_i)\left[1 - (1 - p_i)^{B_i - 1}\right] \quad (10)$$

We solve this using a greedy algorithm, iteratively allocating budget to the samples that yield the highest marginal utility.

**Inference Routing for Cost-Performance Trade-off**

**Setting and Objective.** We further evaluate $V_0$ during the deployment for inference routing. With the rise of inference-time scaling, real-world deployments often maintain a heterogeneous *Model Fleet* $\Pi$, ranging from lightweight, low-latency models to flagship, high-capability ones. Traditional "one-size-fits-all" strategies, which route every query to the strongest one, fail to balance expenses with performance. Similar to the training setting, we treat each candidate model in fleet as an independent policy $\pi$ and construct a capability context $\mathcal{C}$. Inspired by Avengers-Pro (Zhang et al., 2025a), we incorporate both performance $r \in \{0, 1\}$ and normalized cost $\tilde{c} \in [0, 1]$ (derived from model size and token usage) into the context. We define a cost-weighted label as: $r^\beta = \beta r + (1 - \beta)(1 - \tilde{c})$. Consequently, the context is dynamically constructed as $\mathcal{C}_\pi^\beta = \{(x_j, \text{Score}_{\beta,j})\}_{j=1}^N$ based on the cost trade-off preference $\beta$. For instance, a lower $\beta$ prioritizes cost reduction, assigning higher preference to

weaker but cheaper models within the context. The routing decision is then formalized as:

$$\pi^* = \underset{\pi \in \Pi}{\arg\max} \quad V_0\left(x, \mathcal{C}_\pi^\beta\right) \quad (11)$$

We augment our training data with the Open-Reasoner-Zero 57k dataset (Hu et al., 2025). By training three architectures under the same protocol, we harvest over 200k interaction samples, integrating them to heighten the sensitivity to routing dynamics. We also maintain the strict separation that all context samples are drawn from this pre-inferred pool and are disjoint from evaluation queries. The primary advantage of $V_0$ is the *zero-shot generalization*: it adapts to new models added to the fleet or changes in pricing strategies solely by updating the context, without requiring any parameter updates. This allows $V_0$ to flexibly navigate the Pareto frontier between performance and cost.

**Main Results: Stability and Tracking Efficiency.** We first evaluate the ability of $V_0$ to track an evolving policy during GRPO training. As shown in Table 1 and Figure 3, $V_0$ consistently outperforms coupled baselines across all architectures. Notably, it achieves performance parity with the computationally expensive *Step-wise Retrain* method, proving that in-context capability recognition is a viable alternative to frequent retraining. Furthermore, while the Vanilla VM exhibits significant lag and instability due to the coupling dilemma, $V_0$ maintains high accuracy (Intra-AUC > 0.85) from the very first step, effectively tracking policy updates without gradient adaptation.

**Robust Generalization across Distribution Shifts.** We examine whether $V_0$ captures generalizable meta-knowledge

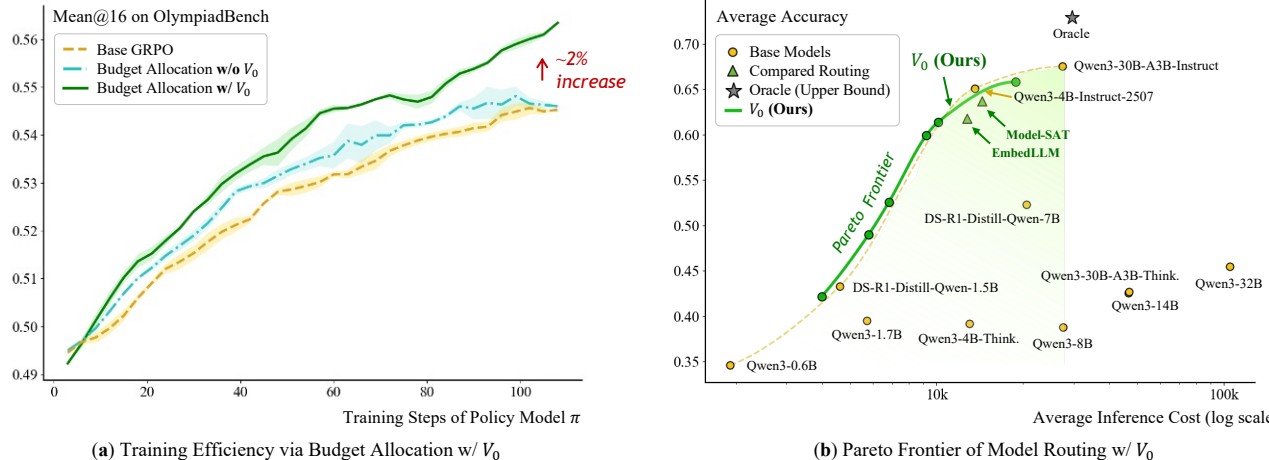

**(a)** Training Efficiency via Budget Allocation w/ $V_0$      **(b)** Pareto Frontier of Model Routing w/ $V_0$

*Figure 5.* **Applications of $V_0$ in Resource Scheduling**. **(a)** By leveraging $V_0$ for step-wise capability estimation, Deploying $V_0$ improves upon GRPO and standard budget allocation baseline without $V_0$ (see Sec. 5 and Equation 10 for more details) on OlympiadBench of Qwen3-4B-Instruct-2507. **(b)** $V_0$ establishes a Pareto frontier between average accuracy and inference cost across 12 benchmarks, outperforming competitive routing baselines such as EmbedLLM and Model-SAT. Please refer to Appendix H for more details.

*Table 3.* **Ablation Studies on $V_0$ Architecture, Training Objectives, and Tuning Strategies.** We compare different connector designs, loss combinations, and tuning parts. The *Overfit Point* indicates the training step where validation performance peaks.

*Table 4.* Impact of Connector Architecture

| Connector Type | Intra AUC | Pair. Acc. | Complex. | Overfit Point |
|---|---|---|---|---|
| Only `last_token` | .621 | .779 | Fast | - |
| MLP | .618 | .835 | Heavy | 4 |
| Cascaded | .585 | .837 | Heavy | 9 |
| MultiScale | .589 | **.861** | Ex. Heavy | 5 |
| Fixed Query | .674 | .782 | Moderate | 25 |
| Pyramidal Fixed | .653 | .805 | Heavy | 22 |
| **Residual Dynamic Query** | **.705** | .839 | Moderate | 39 |

*Table 5.* On Loss Functions

| Loss Type | Intra AUC | Pair. Acc. |
|---|---|---|
| Only Soft CE | .686 | .783 |
| Pairwise (Intra) | .578 | .611 |
| Pairwise (Intra + Inter) | .621 | **.848** |
| **Pairwise (Intra) + Soft CE** | **.705** | .839 |

*Table 6.* On Tuning Modules

| Configuration | TabPFN Head | Intra AUC | Pair. Acc. |
|---|---|---|---|
| w/o Connector | Freeze | .621 | .779 |
| (`last_token`) | Tune | .648 | .813 |
| **Tune Connector** | **Freeze** | **.705** | **.839** |
| | Tune | .594 | .822 |

or merely memorizes prompts. (**1**) **Zero-Shot Transfer**: In Table 2, where test query IDs are strictly excluded from training history, the Vanilla VM collapses to near-random guessing (AUC .560). In contrast, $V_0$ retains robust predictive power (AUC .710), demonstrating its ability to infer capabilities on unseen samples. (**2**) **Multi-Dimensional Robustness**: In Figure 4, whether facing temporal shifts (Early *vs.* Late Steps), architectural changes (Weak *vs.* Strong Arch.), or domain variations (Base/Math (DAPO-Math) to Harder (AIME-24 & AIME-25) or General (GPQA-Diamond)), $V_0$ maintains stable AUCs.

**Applications in Resource Scheduling.** We demonstrate the practical utility of $V_0$ in two scenarios (Figure 5):

- **Dynamic Budget Allocation:** By using $V_0$ to estimate sample difficulty in real-time, we optimize budget allocation during training, accelerating convergence and improving performance on OlympiadBench by $\sim 2\%$ compared to the allocation baselines w/o using $V_0$.
- **Inference Routing:** $V_0$ establishes a Pareto frontier between accuracy and cost. It enables cost-efficient routing strategies that outperform methods like Em-

bedLLM (Zhuang et al., 2025), Model-SAT (Zhang et al., 2025c), allowing for the deployment of smaller models without significant accuracy loss.

**Ablation Studies.**

1. **Architecture** (Table 4): The *Residual Dynamic Query* connector achieves the best performance (AUC .705), offering better generalization than other variants that may be prone to overfitting.
2. **Loss Function** (Table 5): Relying solely on $\mathcal{L}_{rank}$ leads to overfitting (AUC .578), whereas using only $\mathcal{L}_{CE}$ results in poor separability despite achieving good calibration. The design of $V_0$ balances these objectives, ensuring discrimination while maintaining calibration.
3. **Tuning Strategy** (Table 6): Jointly fine-tuning the TabPFN inference head increases overfitting risk. Freezing the pre-trained TabPFN head and tuning only the *Connector* yields optimal results.

## 6. Conclusion

In this paper, we introduce $V_0$, a foundational Generalist Value Model resolving the inherent value model cou-

pling dilemma by reframing value estimation as context-conditional prediction. Synthesizing high-dimensional semantic perception with structured probabilistic reasoning, $V_0$ achieves robust zero-gradient adaptation, enabling accurate policy tracking and dynamic resource scheduling optimization without synchronous training instability. $V_0$ establishes the first pre-training paradigm for capability recognition, showing that model potential is inferable from historical behavior rather than just parameters. While currently focusing on coarse-grained value estimation at state zero ($s_0$), future work will extend this mechanism to the token-level for fine-grained process supervision.

## Acknowledgments

This work was supported by the National Key R&D Program of China (2024YFE0202800), the National Natural Science Foundation of China (NSFC) (62522605 and 62376118), the Basic Research Program of Jiangsu (BK20253021), the Fundamental and Interdisciplinary Disciplines Breakthrough Plan of the Ministry of Education of China (JYB2025XDXM118), the `111 Center` (B26023), and the Collaborative Innovation Center of Novel Software Technology and Industrialization.

## Impact Statement

This paper presents work whose goal is to advance the field of Machine Learning. There are many potential societal consequences of our work, none which we feel must be specifically highlighted here.

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

# Appendix

## A. Theoretical Analysis of Shortcut Learning and Ranking Invariance (section 4)

In this section, we provide the mathematical derivations for the claims made in the main paper regarding the shortcut learning phenomenon and the effectiveness of the pairwise ranking objective.

Let $X \in \mathcal{X}$ denote the input query (prompt), and $Y \in \{0, 1\}$ denote the binary outcome (reward), where 1 represents success. Let $\mathcal{C} \in \mathfrak{C}$ represent the context, defined as the set of historical query-performance pairs for a specific policy. We denote the Shannon entropy of a random variable $Z$ as $H(Z)$ and the conditional entropy as $H(Z \mid W)$. The binary entropy function is denoted as $\mathcal{H}_b(p) = -p \log p - (1-p) \log(1-p)$.

### A.1. Information Capacity Bound

**Proposition A.1.** *For any predictive model attempting to infer $Y$ from features $(X, \mathcal{C})$, the mutual information $I(Y; X, \mathcal{C})$ is bounded by the marginal entropy of the labels $H(Y)$. This upper bound is maximized if and only if the global label distribution is balanced, i.e., $P(Y = 1) = 0.5$.*

*Proof.* By the definition of Mutual Information:

$$I(Y; X, \mathcal{C}) = H(Y) - H(Y \mid X, \mathcal{C}) \tag{12}$$

Since entropy is non-negative, $H(Y \mid X, \mathcal{C}) \geq 0$, leading to the natural upper bound $I(Y; X, \mathcal{C}) \leq H(Y)$.

Let $p = P(Y = 1)$ be the global success rate. The marginal entropy is given by the binary entropy function $f(p) = \mathcal{H}_b(p)$. The first and second derivatives with respect to $p$ are:

$$f'(p) = \log(1-p) - \log(p)$$
$$f''(p) = -\frac{1}{p(1-p)}$$

Since $f''(p) < 0$ for all $p \in (0, 1)$, $\mathcal{H}_b(p)$ is strictly concave. Setting $f'(p) = 0$ yields $p = 0.5$. Thus, the information capacity is maximized uniquely at the balanced distribution. $\square$

### A.2. Mutual Information Decomposition

**Lemma A.2.** *The total information available for predicting $Y$ decomposes into a context-dependent prior term and a query-dependent interaction term:*

$$I(Y; X, \mathcal{C}) = I(Y; \mathcal{C}) + I(Y; X \mid \mathcal{C}) \tag{13}$$

*Proof.* We apply the Chain Rule for Mutual Information. For random variables $A, B, Z$, $I(A; B, Z) = I(A; Z) + I(A; B \mid Z)$. By setting $A = Y$, $B = X$, and $Z = \mathcal{C}$, we obtain:

$$I(Y; X, \mathcal{C}) = \underbrace{(H(Y) - H(Y \mid \mathcal{C}))}_{\text{Context Shortcut}} + \underbrace{(H(Y \mid \mathcal{C}) - H(Y \mid X, \mathcal{C}))}_{\text{Instance Reasoning}} \tag{14}$$

$\square$

### A.3. Shortcut Learning Existence

Here we prove that if policies have different capability levels (variance in performance), a shortcut exists. A model can reduce loss simply by memorizing the capability of the context $\mathcal{C}$, without looking at the query $X$.

---

**Theorem A.3.** *Let $\mu(\mathcal{C}) \triangleq P(Y = 1 \mid \mathcal{C})$ denote the latent capability prior of context $\mathcal{C}$. If $\mathrm{Var}[\mu(\mathcal{C})] > 0$, then $I(Y; \mathcal{C}) > 0$, and we have:*

$$\min_{\theta} \mathcal{L}_{CE}(P(Y \mid \mathcal{C})) < H(Y).$$

*Thus, a model minimizing $\mathcal{L}_{CE}$ can strictly reduce error by fitting prior $\mu(\mathcal{C})$ alone, independent of the input $X$.*

---

*Proof.* The conditional entropy of $Y$ given $\mathcal{C}$ is the expectation of the entropy of the conditional probabilities:

$$H(Y \mid \mathcal{C}) = \mathbb{E}_{\mathcal{C}} \left[ \mathcal{H}_b(\mu(\mathcal{C})) \right] \tag{15}$$

We are given that $\mathbb{E}_{\mathcal{C}}[\mu(\mathcal{C})] = 0.5$ (global balance) and $\mathrm{Var}[\mu(\mathcal{C})] > 0$. Since $\mathcal{H}_b(p)$ is strictly concave, we apply Jensen's Inequality. For a strictly concave function $f$ and a non-constant random variable $Z$:

$$\mathbb{E}[f(Z)] < f(\mathbb{E}[Z]) \tag{16}$$

Substituting our terms:

$$\mathbb{E}_{\mathcal{C}}[\mathcal{H}_b(\mu(\mathcal{C}))] < \mathcal{H}_b(\mathbb{E}_{\mathcal{C}}[\mu(\mathcal{C})]) = \mathcal{H}_b(0.5) = H(Y) \tag{17}$$

Thus, $H(Y \mid \mathcal{C}) < H(Y)$. By definition, $I(Y;\mathcal{C}) = H(Y) - H(Y \mid \mathcal{C}) > 0$.

**Implication for Optimization:** Consider the Cross-Entropy (CE) loss $\mathcal{L}_{\mathrm{CE}}$.

- A *Marginal Baseline* model predicting $\hat{y} = 0.5$ achieves $\mathcal{L} = H(Y) = 1$ bit.

- A *Shortcut* model predicting $\hat{y} = \mu(\mathcal{C})$ achieves $\mathcal{L} = H(Y \mid \mathcal{C}) < 1$ bit.

Since $I(Y;\mathcal{C})$ represents the *easy* statistical correlation (low frequency, low complexity mapping $C \to [0,1]$) compared to the complex high-dimensional interaction $I(Y; X \mid \mathcal{C})$, gradient descent algorithms will prioritize minimizing the error component associated with $I(Y;\mathcal{C})$, leading to the shortcut solution. $\square$

### A.4. Ranking Loss Invariance

To mitigate the shortcut described in Theorem A.3, we utilize a pairwise ranking loss. A key advantage of this objective is its mathematical invariance to any additive bias derived solely from the context. We distinguish between the *logit score* $s(x,\mathcal{C};\phi) \in \mathbb{R}$ and the final *value probability* $V_0(x,\mathcal{C};\phi) = \sigma(s(x,\mathcal{C};\phi)) \in [0,1]$. The ranking optimization is performed directly on the logit space $\Delta s$.

> **Theorem A.4.** *Let $\tilde{s}(x,\mathcal{C}) = s(x,\mathcal{C}) + b(\mathcal{C})$ be a scoring function perturbed by an arbitrary context-dependent bias $b(\mathcal{C})$. The gradient of the ranking loss with respect to parameters $\phi$ of $V_0$ satisfies:*
>
> $$\nabla_\phi \mathcal{L}_{rank}(\tilde{s}) = \nabla_\phi \mathcal{L}_{rank}(s).$$

*Proof.* Consider a pair of samples $(x_i, x_j)$ drawn from the *same* context $\mathcal{C}$ with opposing labels. The Bradley-Terry probability of $x_i$ being preferred over $x_j$ is modeled using the difference in their logit scores, denoted as $\Delta s_{ij}$:

$$P(x_i \succ x_j \mid \mathcal{C}) = \sigma(\Delta s_{ij}) = \frac{1}{1 + e^{-(s(x_i,\mathcal{C}) - s(x_j,\mathcal{C}))}} \tag{18}$$

When the scoring function is perturbed by a context-specific bias $b(\mathcal{C})$, the perturbed logit difference $\Delta \tilde{s}_{ij}$ becomes:

$$\begin{aligned}
\Delta \tilde{s}_{ij} &= \tilde{s}(x_i,\mathcal{C}) - \tilde{s}(x_j,\mathcal{C}) \tag{19} \\
&= [s(x_i,\mathcal{C};\phi) + b(\mathcal{C})] - [s(x_j,\mathcal{C};\phi) + b(\mathcal{C})] \tag{20} \\
&= s(x_i,\mathcal{C};\phi) - s(x_j,\mathcal{C};\phi) \tag{21} \\
&= \Delta s_{ij} \tag{22}
\end{aligned}$$

The bias term $b(\mathcal{C})$ is eliminated algebraically during the subtraction. The ranking loss is defined as $\mathcal{L}_{\mathrm{rank}} = -\log \sigma(\Delta s_{ij})$. The gradient with respect to the model parameters $\phi$ is:

$$\nabla_\phi \mathcal{L}_{\mathrm{rank}} = (\sigma(\Delta s_{ij}) - 1) \cdot \nabla_\phi (s(x_i,\mathcal{C};\phi) - s(x_j,\mathcal{C};\phi)) \tag{23}$$

Since $\Delta s_{ij}$ is independent of $b(\mathcal{C})$, both the scalar loss value and the gradient vector remain unaffected by shifts in the context prior. Consequently, minimizing $\mathcal{L}_{\mathrm{rank}}$ forces the model to extract features that discriminate $x_i$ from $x_j$ *within* the context, effectively maximizing the conditional mutual information $I(Y; X \mid \mathcal{C})$. $\square$

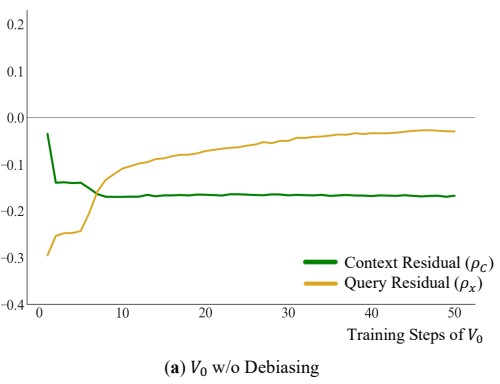

(a) $V_0$ w/o Debiasing

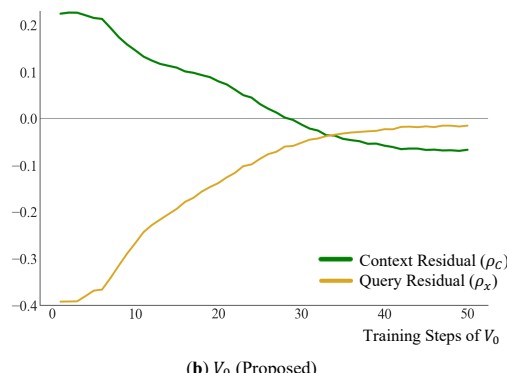

(b) $V_0$ (Proposed)

*Figure 6.* **Evolution of Residual Orthogonality during Training.** The *left* plot shows the $V_0$ fine-tuning TabPFN head, where residuals fail to converge to zero, indicating the overfitting. The *right* plot demonstrates our $V_0$, where both *Context Residual* and *Query Residual* converge towards zero (dashed line), confirming that the model has successfully decoupled reasoning from statistical shortcuts.

# B. Empirical Analysis Framework: Residual Orthogonality

In subsection 4.3, we theoretically decomposed the mutual information into a *shortcut term* $I(Y;\mathcal{C})$ and a *reasoning term* $I(Y;X \mid \mathcal{C})$. A critical challenge in training generalist value models is distinguishing whether the model is learning robust causal reasoning (estimating $P(Y \mid X,\mathcal{C})$) or merely memorizing context shortcuts (estimating $P(Y \mid \mathcal{C})$). To rigorously verify the effectiveness of our debiasing strategy, we establish a diagnostic framework based on **Residual Orthogonality**. This framework analyzes the statistical properties of the model's errors relative to the known capabilities of the policies.

## B.1. Defining Statistical Priors

To quantify the shortcut information available to the model, we calculate the empirical priors from the training history.

**Definition B.1** (Context Prior $\mu(\mathcal{C})$)**.** We operationalize the theoretical latent capability $\mu(\mathcal{C})$ as the average empirical success rate of a specific policy checkpoint $\mathcal{C}$ across all its historical samples:

$$\mu(\mathcal{C}) \triangleq \frac{1}{N_{\mathcal{C}}} \sum_{j=1}^{N_{\mathcal{C}}} y_j^{(\mathcal{C})} \tag{24}$$

where $N_{\mathcal{C}}$ denotes the total number of historical rollout samples available for policy $\mathcal{C}$.

- **Significance:** This term serves as a proxy for the *Shortcut Information*. It represents the policy's baseline strength (*e.g.*, a 70B model has a higher $\mu(\mathcal{C})$ than a 7B model). A model relying solely on this prior acts as a heuristic lookup table, ignoring the specific semantic complexity of the query.

**Definition B.2** (Query Difficulty $D_x$)**.** For a given query $x$, the Query Difficulty prior $D_x$ is the average success rate on this query across all policies that have attempted it:

$$D_x \triangleq \frac{1}{M_x} \sum_{k=1}^{M_x} y_k^{(x)} \tag{25}$$

where $M_x$ is the frequency that $x$ has been evaluated. This term proxies the intrinsic difficulty independent of the model.

## B.2. Diagnostic Metric: Residual Orthogonality

To empirically validate the Shift-Invariance property, we focus on the correlation between the model's prediction errors and the statistical priors defined above. We establish a dual-metric diagnostic framework:

1. **Context Residual:** Measures dependency on model identity.

$$\text{Residual}_{\mathcal{C}} = \text{Spearman } \rho(\underbrace{\hat{y} - y}_{\text{Error}}, \mu(\mathcal{C})) \tag{26}$$

2. **Query Residual:** Measures dependency on statistical difficulty.

$$\text{Residual}_x = \text{Spearman } \rho(\underbrace{\hat{y} - y}_{\text{Error}}, D_x) \tag{27}$$

where $\hat{y}$ is the predicted value, $y$ is the ground truth label, and $\rho$ denotes the rank correlation coefficient.

### B.3. Interpretation: Acquisition *vs*. Dependency (Figure 6)

A common misconception is that a debiased model should ignore the inherent capabilities of different policies or the difficulty of queries. On the contrary, these priors are valid components of the ground truth. Our framework distinguishes between valid **information acquisition** and invalid **shortcut dependency** through the lens of residual orthogonality.

We posit that a perfectly trained value model $V^*$ decomposes into the baseline priors and a reasoning delta:

$$V^*(x, \mathcal{C}) = \underbrace{\Phi(\mu(\mathcal{C}), D_x)}_{\text{Statistical Baselines}} + \underbrace{\Delta(x, \mathcal{C})}_{\text{Reasoning Interaction}} \tag{28}$$

Consequently, when both trajectories settle near zero, it demonstrates that the model's remaining errors are irreducible random noise ($\epsilon$), statistically independent of both the policy's history and the query's popularity. This confirms the model has transitioned from shortcut learning to robust causal reasoning.

## C. Derivation of Budget Allocation Utility (section 5)

In this section, we provide the theoretical justification for the utility function used in the Budget Allocation module (Equation 10 in the main text). We first establish the relationship between the gradient norm and the sum of absolute advantages in GRPO. We then derive the closed-form approximation for the expected utility.

### C.1. Gradient Bound via Advantage Sum

We define the utility of a set of rollouts based on the potential magnitude of the gradient update. We propose that the norm of the policy gradient vector is bounded by the sum of the absolute advantages within the group.

**Proposition C.1.** *In Group Relative Policy Optimization (GRPO), let $J(\theta)$ be the objective function. The norm of the gradient update is bounded by the sum of absolute advantages, scaled by the Lipschitz constant of the policy's log-likelihood:*

$$||\nabla_\theta J(\theta)|| \leq \gamma(s) \cdot \frac{1}{G} \sum_{i=1}^{G} |A_i| \tag{29}$$

*where $G$ is the group size, $A_i$ is the advantage of the $i$-th sample, and $\gamma(s)$ is a context-dependent constant.*

*Proof.* The gradient of the GRPO objective with respect to parameters $\theta$ is given by:

$$\nabla_\theta J(\theta) = \mathbb{E}\left[\frac{1}{G}\sum_{i=1}^{G} A_i \cdot \nabla_\theta \log \pi_\theta(y_i|x)\right] \tag{30}$$

We assume the policy model $\pi_\theta$ satisfies a Lipschitz continuity condition locally, such that the norm of the score function is bounded for a given input $x$:

$$||\nabla_\theta \log \pi_\theta(y|x)|| \leq \gamma(x; \theta) \tag{31}$$

Applying the norm to the gradient estimator and utilizing the Triangle Inequality ($|| \sum v_i || \le \sum ||v_i||$):

$$||\nabla_\theta J(\theta)|| = \left\Vert \frac{1}{G} \sum_{i=1}^{G} A_i \cdot \nabla_\theta \log \pi_\theta(y_i|x) \right\Vert \tag{32}$$

$$\le \frac{1}{G} \sum_{i=1}^{G} ||A_i \cdot \nabla_\theta \log \pi_\theta(y_i|x)|| \tag{33}$$

$$= \frac{1}{G} \sum_{i=1}^{G} |A_i| \cdot ||\nabla_\theta \log \pi_\theta(y_i|x)|| \tag{34}$$

Substituting the Lipschitz bound $\gamma(x; \theta)$:

$$||\nabla_\theta J(\theta)|| \le \frac{\gamma(x; \theta)}{G} \sum_{i=1}^{G} |A_i| \tag{35}$$

$\square$

This result suggests that maximizing the sum of absolute advantages $\sum |A_i|$ effectively maximizes the upper bound of the update *force* (gradient magnitude), maximizing the potential learning signal for a given step.

### C.2. Derivation of Expected Signal Strength

We now derive the closed-form utility function. Let $B$ denote the group size (referred to as budget in Equation 10) and $p$ denote the predicted success probability of the policy for a given prompt.

Let the random variable $k_{\text{succ}} \sim \text{Binomial}(B, p)$ represent the number of correct responses in a group of size $B$. The probability of observing exactly $k$ correct responses is:

$$P(k_{\text{succ}} = k|B, p) = \binom{B}{k} p^k (1-p)^{B-k} \tag{36}$$

In GRPO with binary rewards ($r \in \{0, 1\}$), for a group with $k$ successes, the group mean is $\mu = k/B$. We have:

$$\sigma = \sqrt{\frac{k(B-k)}{B^2}} = \frac{\sqrt{k(B-k)}}{B} \tag{37}$$

The advantages for positive samples ($A_{\text{pos}}$) and negative samples ($A_{\text{neg}}$) are calculated as:

$$A_{\text{pos}}(k) = \frac{1-\mu}{\sigma} = \frac{1-k/B}{\frac{1}{B}\sqrt{k(B-k)}} = \sqrt{\frac{B-k}{k}} \tag{38}$$

$$A_{\text{neg}}(k) = \frac{0-\mu}{\sigma} = \frac{-k/B}{\frac{1}{B}\sqrt{k(B-k)}} = -\sqrt{\frac{k}{B-k}} \tag{39}$$

We define the *Gradient Signal Strength* $S(k)$ as the sum of absolute advantages in the group (ignoring the constant factor $1/G$ from the previous part for optimization purposes):

$$
\begin{aligned}
S(k) = \sum |A_i| &= k \cdot |A_{\text{pos}}(k)| + (B-k) \cdot |A_{\text{neg}}(k)| \\
&= k\sqrt{\frac{B-k}{k}} + (B-k)\sqrt{\frac{k}{B-k}} \\
&= \sqrt{k(B-k)} + \sqrt{k(B-k)} = 2\sqrt{k(B-k)}
\end{aligned} \tag{40}
$$

The expected signal strength is the expectation over the binomial distribution, summing over valid cases where variance is non-zero (*i.e.*, $k \ne 0$ and $k \ne B$). We have:

$$\mathbb{E}[S] = \sum_{k=1}^{B-1} P(k_{\text{succ}} = k) \cdot 2\sqrt{k(B-k)} \tag{41}$$

Calculating this involves fractional moments and is computationally expensive to solve in closed form for real-time allocation. To obtain a tractable surrogate, we introduce a scaling factor $\lambda(k)$ based on the positive advantage:

$$\lambda(k) \triangleq \frac{1}{2}A_{\text{pos}}(k) = \frac{1}{2}\sqrt{\frac{B-k}{k}} \tag{42}$$

Multiplying the signal strength by this factor allows us to approximate the utility by focusing on the contribution of negative samples (the learning from mistakes signal):

$$S_{\text{proxy}}(k) = S(k) \cdot \lambda(k) = 2\sqrt{k(B-k)} \cdot \frac{1}{2}\sqrt{\frac{B-k}{k}} = B - k \tag{43}$$

We now calculate the expected utility of this proxy metric, effectively measuring the expected number of errors conditioned on the gradients being non-zero:

$$\text{Utility}(B, p) \approx \mathbb{E}[S_{\text{proxy}}] = \sum_{k=1}^{B-1} P(k_{\text{succ}} = k) \cdot (B - k) \tag{44}$$

This summation can be solved by considering the full binomial expectation and subtracting the edge cases:

$$\sum_{k=1}^{B-1} P(k)(B-k) = \underbrace{\sum_{k=0}^{B} P(k)(B-k)}_{\mathbb{E}[B-k_{\text{succ}}]} - \underbrace{P(0)(B-0)}_{\text{All Wrong}} - \underbrace{P(B)(B-B)}_{\text{All Correct}} \tag{45}$$

$$= (B - \mathbb{E}[k_{\text{succ}}]) - B(1-p)^B - 0 \tag{46}$$

$$= B(1-p)\left[1 - (1-p)^{B-1}\right] \tag{47}$$

Finally, matching the Equation 10 of the main paper:

$$\text{Utility}(B_i, p_i) = B_i(1-p_i)\left[1 - (1-p_i)^{B_i-1}\right] \tag{48}$$

## D. Information-Theoretic Interpretation (section 5)

In the main paper (subsection 4.3), we decomposed the information available for value estimation into a context prior term and a causal interaction term:

$$I(Y; X, \mathcal{C}) = \underbrace{I(Y; \mathcal{C})}_{\text{Context Shortcut}} + \underbrace{I(Y; X \mid \mathcal{C})}_{\text{Causal Reasoning}} \tag{49}$$

In this section, we first analyze the Reward Model & $k$NN-Contextual baselines used in our experiments through this information-theoretic lens. We demonstrate that these baselines represent incomplete approximations of the full mutual information term, whereas $V_0$ aims to capture the complete conditional distribution.

### D.1. Reward Models as Estimators of Global Difficulty $I(Y; X)$

The *Reward Model (RM)* baseline (*e.g.*, *Qwen2.5-Math-RM-72B*) estimates the value of a query $x$ using a fixed set of parameters $\theta_{\text{RM}}$, independent of the policy $\pi$ being evaluated. Formally, it models:

$$V_{\text{RM}}(x) \approx P(Y = 1 | X = x, \theta_{\text{RM}}) \tag{50}$$

**Theoretical Limitation:** By ignoring the specific policy context $\mathcal{C}$, the RM assumes conditional independence $Y \perp \mathcal{C} | X$. However, this assumption holds only if all policies have identical capabilities (*i.e.*, $\pi_i \approx \pi_j$). In our setting, where policies evolve or vary in size, this assumption is violated. The information gap is quantified by the conditional mutual information of the context given the query:

$$\text{Information Gap} = I(Y; X, \mathcal{C}) - I(Y; X) = I(Y; \mathcal{C}|X) \tag{51}$$

This implies that the RM cannot distinguish between a *hard query for a weak model* and a *hard query for a strong model*.

**D.2. $k$NN as Non-Parametric Estimation of $I(Y; X|\mathcal{C})$**

The $k$**NN-Contextual** baseline stores the history $\mathcal{C} = \{(x_i, r_i)\}_{i=1}^N$ explicitly. For a target query $x_t$, it retrieves the set of nearest neighbors $\mathcal{N}_k(x_t) \subset \mathcal{C}$ based on semantic similarity and estimates:

$$V_{k\text{NN}}(x_t, \mathcal{C}) = \frac{1}{k} \sum_{(x_j, r_j) \in \mathcal{N}_k(x_t)} r_j \tag{52}$$

**Theoretical Mapping:** Unlike the RM, the $k$NN approach conditions on the context $\mathcal{C}$. It can be viewed as a *non-parametric local estimator* of the interaction term $I(Y; X \mid \mathcal{C})$. It attempts to approximate the posterior predictive distribution $P(Y \mid X, \mathcal{C})$ by assuming that the capability function is locally constant in the semantic embedding space.

**Theoretical Limitation:** While $k$NN theoretically accesses the correct information channel $I(Y; X \mid \mathcal{C})$, it suffers from two critical limitations compared to the parametric $V_0$:

1. The estimation error of $k$NN scales with the sparsity of the context $\mathcal{C}$. In high-dimensional semantic spaces, the nearest neighbors might still be semantically distant, leading to high variance in the estimator.

2. $k$NN relies strictly on geometric proximity. It cannot learn higher-order logical patterns (meta-knowledge). For example, if a policy consistently fails at geometry problems, $V_0$ can infer this capability boundary even if the specific geometry query $x_t$ has no close vector neighbors in $\mathcal{C}$. $k$NN fails to extract these latent capability features, bounded by the density of the observed support.

**D.3. Linking Evaluation Metrics to Information Components**

To validate that $V_0$ learns the robust causal reasoning term $I(Y; X \mid \mathcal{C})$ rather than relying on the context shortcut $I(Y; \mathcal{C})$, we map our evaluation metrics to the information decomposition.

D.3.1. INTRA-CONTEXT AUC AS A PROXY FOR CAUSAL REASONING $I(Y; X \mid \mathcal{C})$

The Intra-Context AUC measures the ability to rank queries $x_i, x_j$ correctly within a single fixed policy checkpoint $\mathcal{C}$. This corresponds to evaluating the discriminative power strictly under the conditional distribution $P(Y \mid X, \mathcal{C} = \mathcal{C}_\pi)$.

**Formal Relation:** Consider a *shortcut-only* estimator $V_{\text{shortcut}}(x, \mathcal{C}) = P(Y = 1 \mid \mathcal{C})$, which captures the entire context prior term $I(Y; \mathcal{C})$ but ignores the query interaction $I(Y; X \mid \mathcal{C})$. For any fixed context $\mathcal{C}$, this estimator outputs a constant score $s = \mu(\mathcal{C})$ for all $x$. Consequently, the ROC curve collapses to the diagonal, yielding an AUC of $0.5$.

Therefore, any gain in *Intra-Context AUC* above 0.5 is attributable *exclusively* to the utilization of the interaction term:

$$\text{Gain}_{\text{AUC}} \propto I(\hat{Y}; X \mid \mathcal{C}) \tag{53}$$

By optimizing the Pairwise Ranking Loss $\mathcal{L}_{\text{rank}}$ (which is shift-invariant), $V_0$ explicitly maximizes this term, ensuring the model learns the causal relationship between query and success, rather than relying on the easier context prior.

D.3.2. PAIRWISE CALIBRATION ACCURACY AND CONTEXT DEPENDENCE $I(Y; \mathcal{C})$

The *Pairwise Calibration Accuracy* evaluates pairs of the *same* query evaluated by *different* policy checkpoints: $(x, \mathcal{C}_i)$ *vs.* $(x, \mathcal{C}_j)$, where the ground truth implies a shift in capability ($r_i \neq r_j$). This metric isolates the variability in $Y$ driven by $\mathcal{C}$ while $X$ remains constant.

**Formal Relation:** Consider a standard *Reward Model* estimator $V_{\text{RM}}(x) \approx P(Y \mid X)$, which captures only $I(Y; X)$. Since $V_{\text{RM}}$ is independent of $\mathcal{C}$, for any query $x$, the predicted score remains invariant across policies:

$$s(x, \mathcal{C}_i) = s(x, \mathcal{C}_j) \implies P(\text{correct ranking}) \approx 0.5 \quad (\text{like random guess}) \tag{54}$$

We distinguish three cases to illustrate why this metric should be interpreted alongside AUC:

1. **Reward Models (Underfitting $\mathcal{C}$):** A standard RM captures only $I(Y; X)$. Since it is independent of $\mathcal{C}$, $s(x, \mathcal{C}_i) = s(x, \mathcal{C}_j)$, leading to random guessing accuracy ($\approx 0.5$).

2. **Shortcut Models (Overfitting $\mathcal{C}$):** A model that learns *only* the shortcut $I(Y; \mathcal{C})$ (*i.e.*, judging policy strength while ignoring the query) will achieve high pairwise accuracy. It correctly identifies that a stronger policy $\mathcal{C}_{\text{strong}}$ is more likely to succeed than $\mathcal{C}_{\text{weak}}$ on average. However, such a model would yield an Intra-Context AUC of 0.5.

3. **Generalist $V_0$ (Balanced):** A robust model must achieve high scores on *both* metrics. High *Pairwise Accuracy* confirms it tracks the non-stationary policy (capturing $I(Y; \mathcal{C})$), while high *Intra-Context AUC* confirms it understands specific problem difficulties (capturing $I(Y; X \mid \mathcal{C})$).

## E. Case Study: The Insufficiency of State-Only Value Estimation

A prevailing hypothesis in recent research suggests that the dependency of the value function on the specific policy $\pi$ can be relaxed, simplifying $V^\pi(s)$ to a policy-agnostic $V(s)$. The rationale is that the partial trajectory $s$ generated by a LLM already implicitly encodes sufficient information about the capabilities. However, we present a counter-example that challenges this claim. We observe that models with vastly different capabilities (*e.g.*, a 1.5B model *vs.* a 4B model) often generate semantically identical prefixes for mathematical reasoning tasks due to the deterministic nature of initial logical deductions. Yet, their final outcomes diverge significantly based on their varying reasoning depths. This phenomenon proves that $V(s)$ is insufficient and that explicit conditioning on the policy $\pi$ (or its capability context $\mathcal{C}_\pi$, as in $V_0$) is essential.

**The Counter-Example Scenario.** Consider the following complex analysis problem:

> **Prompt:** Given a complex number $z$ such that $z - \frac{4}{z}$ is purely imaginary, find the integer approximation of the minimum value of $|z - 1 - i|$.

We compare the rollout trajectories of a 1.5B Model (Weak Policy) and a 4B Model (Strong Policy). The 4B model correctly solves this problem, while the 1.5B model fails.

**Phase 1: Indistinguishable Initial Trajectories.** In the initial modeling and derivation phase, the generated trajectories are nearly identical. This is because the mathematical derivation follows a rigorous, almost unique path:

- Both models begin by letting $z = x + yi$ (or $a + bi$).
- Both models calculate $z - \frac{4}{z}$ and simplify it to separate the real and imaginary parts.
- Both models apply the purely imaginary condition (setting real part to 0) and correctly derive the two possible loci for $z$:
    1. $x = 0$ (The imaginary axis).
    2. $x^2 + y^2 = 4$ (A circle centered at the origin with radius 2).

At this stage, a state-only value model $V(s)$ may assign the same value to both trajectories, as the text $s$ is mathematically identical and correct.

**Phase 2: Divergence Driven by Intrinsic Capability.** The divergence occurs immediately after determining the locus, specifically during the optimization step: "How to minimize the distance from the circle $x^2 + y^2 = 4$ to the point $(1, 1)$."

- **4B Model (Success):** The model employs a geometric approach. It calculates the modulus of the target point $|1 + i| = \sqrt{2}$. Since $\sqrt{2} < 2$, it recognizes the point is *inside* the circle. It then correctly deduces that the minimum distance is along the radius: $R - |1 + i| = 2 - \sqrt{2} \approx 0.58$, leading to the correct integer approximation.
- **1.5B Model (Failure):** The model hesitates on the geometric relationship (unable to robustly determine if the point is inside or outside). It abandons the geometric insight and switches to an algebraic/trigonometric substitution method, setting $x = 2\cos\theta, y = 2\sin\theta$. Due to the complexity of minimizing the resulting trigonometric function, the model hallucinates intermediate steps or makes calculation errors, leading to an incorrect result.

This case study demonstrates that for two policies $\pi_{\text{weak}}$ and $\pi_{\text{strong}}$, we can have a state $s$ such that $s_{\pi_{\text{weak}}} = s_{\pi_{\text{strong}}}$, yet the true values differ drastically: $V^{\pi_{\text{weak}}}(s) \approx 0$ while $V^{\pi_{\text{strong}}}(s) \approx 1$. Therefore, the value function cannot be decoupled from the policy. The indistinguishability of early trajectories necessitates an explicit representation of the policy's capability, validating the design of $V_0(\pi, s)$ where $\pi$ is provided via context.

# F. Detailed Implementation, Hyperparameters, and Computational Analysis

**Residual Query Adapter Configuration.** We configure the Residual Query Adapter with $N_{\text{static}} = 168$ static queries and a projection dimension of $= 6$. We select these specific values for: The total feature dimension yields $168 \times 6 = 1008$, which approximates 1k; The projection dimension of 6 allows for distinct differentiation between queries while ensuring divisibility by 3, which aligns with the encoding structure of the TabPFN inference head (processing features in groups of 3).

## F.1. Budget Allocation Settings

For the Dynamic Budget Allocation experiments (section 5), we integrated $V_0$ into the GRPO training loop. The specific hyperparameters used for the policy optimization are detailed in Table 7.

*Table 7.* Hyperparameters for GRPO Training with $V_0$-guided Budget Allocation.

| Hyperparameter | Value |
|---|---|
| Learning Rate | $1 \times 10^{-6}$ |
| KL Coefficient (`kl_loss_coef`) | 0.001 |
| Global Batch Size | 512 |
| PPO Mini-Batch Size | 256 |
| Group Size ($G$) | 16 |
| PPO Clip Ratio (Low) | 0.2 |
| PPO Clip Ratio (High) | 0.28 |
| Training Steps | 132 |

**Allocation Constraints.** When $V_0$ dynamically allocates the rollout budget $B_i$ for a specific prompt $x_i$, we enforce hard constraints to prevent computational collapse or explosion. The allocated budget is clipped to the range:

$$B_i \in [2, 128] \tag{55}$$

## F.2. Inference Routing Configuration

**Model Fleet and Benchmarks.** We construct a heterogeneous model fleet consisting of 11 widely-used open-source LLMs, with parameter counts ranging from 0.6B to 32B. To ensure a robust evaluation of capability, the fleet is evaluated across a comprehensive suite of 12 benchmarks covering mathematics, logical reasoning, and general knowledge. The performance metric reported is the *Average* `avg@10` (success rate averaged over 10 stochastic generations).

**Cost Formulation.** To simulate real-world API pricing or serving costs, we calculate the *Inference Cost* for each model based on token consumption and model size. The cost $c_\pi$ for model $\pi$ is defined as:

$$c_\pi = \text{Params Ratio of } \pi \times \text{Tokens}_{\text{avg.}} \tag{56}$$

*e.g.*, the 7B model's ratio is 7, the 30B-A3B model's ratio is 15.

**Baselines and Pareto Analysis.** We compare $V_0$ against two competitive routing baselines: EmbedLLM and Model-SAT. We also include an Oracle baseline, representing the theoretical upper bound where the router perfectly selects the most efficient model that can solve the query.

For our method ($V_0$), we generate the Pareto frontier by sweeping the cost-tradeoff coefficient $\beta$. This allows us to visualize the transition from *maximum efficiency* modes (prioritizing low cost) to *maximum performance* modes (prioritizing accuracy).

## F.3. Data Construction and Ground Truth

To train $V_0$ and evaluate GT performance, we generate rollouts for every query-policy pair. The hyperparameters are:

## F.4. Computational Efficiency Analysis

A critical requirement for an auxiliary value model is that it should not introduce significant latency. $V_0$ is designed to be lightweight. On a standard inference setup, $V_0$ processes a batch of 8 samples in approximately `600ms`. This low overhead confirms that $V_0$ is viable for real-time routing and dynamic training allocation without becoming a bottleneck.

*Table 8.* Hyperparameters for Rollouts for Every Query-policy Pair.

| Hyperparameter | Value |
|---|---|
| Sampling $n$ | 10 (and we compute `avg@10`) |
| Temperature | 1.0 |
| Top-p | 0.9 |

## G. Context Length Scaling Analysis

We conduct an ablation study on the context size $N$. The results are presented in Table 9.

*Table 9.* Impact of Context Size on Value Estimation Performance. We evaluate $V_0$ varying the number of historical query-performance pairs $N$ in the context $\mathcal{C}_\pi$, following the experimental setup of Table 3.

| Context Size ($N$) | Intra AUC | Pair. Acc. |
|---|---|---|
| 32 | 0.538 | 0.765 |
| 64 | 0.553 | 0.776 |
| 128 | 0.589 | 0.804 |
| 256 | 0.705 | 0.839 |
| 512 | 0.733 | 0.856 |

**Discussion.** We observe a distinct performance threshold regarding the context length. With small contexts ($N \leq 128$), the Intra-Context AUC remains constrained near 0.5, indicating that the model fails to discriminate query difficulty effectively. This suggests that a limited sample size is statistically insufficient to characterize the complex latent capability of a policy. Performance improves significantly at $N = 256$, as the context becomes dense enough to represent the policy's identity.

## H. Details of Figure 5

To further validate the effectiveness of $V_0$ as a resource scheduler, we provide a fine-grained analysis of the Budget Allocation experiment across five distinct mathematical benchmarks: AIME 2024, AIME 2025, AMC 23, MATH 500, and OlympiadBench. As discussed in section 5, the "Budget Allocation w/o $V_0$" baseline applies Equation 10 and relies on lagged heuristics (success rates from previous epochs), whereas "Budget Allocation w $V_0$" applies Equation 10 but utilizes $V_0$ to predict the probability of the current policy on specific prompts in real-time, allowing for zero-shot budget assignment.

The detailed results are presented in Table 11. We observe that Budget Allocation w/ $V_0$ consistently outperforms the standard GRPO baseline across all benchmarks and surpasses the heuristic allocation method (w/o $V_0$) on four out of five datasets. Notably, on the highly challenging AIME 2024 benchmark, $V_0$ achieves a significant improvement, increasing accuracy from 41.04% (GRPO) and 44.58% (w/o $V_0$) to 50.21%. This indicates that $V_0$'s ability to identify the capability boundaries of the model is particularly beneficial for difficult tasks where effectively allocating compute to solvable but hard problems yields the highest marginal utility. While the heuristic method performs marginally better on MATH 500 (+0.47%), $V_0$ demonstrates superior generalization and efficiency on complex reasoning tasks, confirming its robustness as a dynamic budget allocator.

We also show the details of the model fleet for Inference Routing in Table 10.

*Table 10.* **Detailed Statistics of the Model Fleet.** We report the average accuracy across 12 benchmarks and the corresponding average inference cost. The *Base Models* section lists the candidate models available in the fleet, while the subsequent sections show the aggregate performance of different routing strategies.

| Model | addsub | aime 2024 | aime 2025 | amc23 | college math | gaokao 2023en | gaokao math cloze | gpqa | math hard | math500 | minerva math | olympiad | avg. perf | avg. cost |
|---|---|---|---|---|---|---|---|---|---|---|---|---|---|---|
| DeepSeek-R1-Distill-Qwen-1.5B | .911 | .093 | .113 | .403 | .566 | .596 | .692 | .120 | .448 | .657 | .283 | .315 | .433 | 4,605 |
| DeepSeek-R1-Distill-Qwen-7B | .953 | .163 | .173 | .523 | .632 | .676 | .807 | .216 | .572 | .742 | .426 | .399 | .523 | 20,535 |
| Qwen3-0.6B | .899 | .007 | .040 | .275 | .507 | .522 | .463 | .110 | .296 | .558 | .230 | .251 | .346 | 1,911 |
| Qwen3-1.7B | .960 | .033 | .070 | .338 | .550 | .561 | .566 | .107 | .338 | .610 | .341 | .268 | .395 | 5,717 |
| Qwen3-14B | .968 | .063 | .047 | .377 | .572 | .594 | .567 | .201 | .378 | .655 | .399 | .295 | .426 | 46,484 |
| Qwen3-30B-A3B-Instruct-2507 | .989 | .343 | .307 | .710 | .715 | .818 | .931 | .469 | .791 | .884 | .569 | .574 | .675 | 27,428 |
| Qwen3-30B-A3B-Thinking-2507 | .976 | .007 | .010 | .230 | .655 | .593 | .848 | .267 | .285 | .543 | .514 | .192 | .427 | 46,786 |
| Qwen3-32B | .979 | .100 | .060 | .392 | .582 | .618 | .621 | .260 | .420 | .667 | .429 | .327 | .455 | 105,111 |
| Qwen3-4B-Instruct-2507 | .972 | .287 | .270 | .740 | .695 | .807 | .922 | .418 | .766 | .856 | .510 | .568 | .651 | 13,577 |
| Qwen3-4B-Thinking-2507 | .936 | .020 | .010 | .243 | .622 | .567 | .763 | .180 | .256 | .520 | .406 | .175 | .392 | 12,983 |
| Qwen3-8B | .976 | .027 | .020 | .330 | .531 | .555 | .528 | .161 | .325 | .603 | .341 | .256 | .388 | 27,523 |

*Table 11.* **Detailed Performance Comparison of Budget Allocation on Qwen3-4B-Instruct-2507.** We report the accuracy across five benchmarks. Budget Allocation w/o $V_0$ utilizes lagged heuristics from previous training epochs, while Budget Allocation w/ $V_0$ uses our proposed generalist value model for real-time difficulty estimation. The best performance in each column is highlighted in **bold**.

| Method | AIME 2024 | AIME 2025 | AMC 23 | MATH 500 | OlympiadBench |
|---|---|---|---|---|---|
| GRPO | .4104 | .3188 | .8578 | .8931 | .5453 |
| Budget Allocation w/o $V_0$ | .4458 | .3604 | .8703 | **.9088** | .5497 |
| Budget Allocation w/ $V_0$ (Ours) | **.5021** | **.3656** | **.8984** | .9041 | **.5634** |

# I. Extended Related Work

**Limitations of Current Optimization Paradigms.** Current paradigms for post-training optimization can be broadly categorized into value-free and coupled approaches, each presenting distinct challenges. Value-free methods, such as Group Relative Policy Optimization (GRPO), are designed to circumvent the architectural complexity of maintaining an independent value model. However, they frequently encounter severe bias-variance tradeoffs, particularly when applied to complex reasoning tasks (Zheng et al., 2023; Fan et al., 2025). To mitigate the high variance associated with gradient estimation, these methods typically necessitate massive sampling budgets (Hu et al., 2025; Liu et al., 2025). Yet, this reliance on extensive sampling faces a critical failure mode: on tasks that are either trivial (fully mastered) or effectively impossible (unsolvable), the reward variance vanishes. This phenomenon leads to *advantage collapse*, rendering the optimization signal ineffective (Liu et al., 2025). Furthermore, the significant variability in rollout lengths makes such large-scale sampling not only computationally prohibitive but also a source of severe training instability (Yuan et al., 2025). Conversely, coupled value models, such as Proximal Policy Optimization (PPO), offer a theoretical advantage by explicitly estimating expected returns. However, they struggle with a fundamental coupling dilemma. Because the value function relies on the policy's parameters, mechanisms for synchronizing them (Yue et al., 2025; Chen et al., 2024a; Liu et al., 2024) must continuously track a non-stationary target. This requirement forces the value model to adapt to rapid distribution shifts, a process that is computationally expensive and frequently triggers training oscillations (Han et al., 2024).

**Budget Allocation and Efficient Sampling in RL.** Recent advancements in RL for LLMs have pivoted from uniform sampling to dynamic budget allocation, striving to maximize gradient efficiency within computational limits. This body of research primarily optimizes allocation based on *problem difficulty* and *learning potential*. Zeng et al. (2025) provide a theoretical foundation by establishing that the optimal rollout quantity should be proportional to gradient variance, effectively directing the budget toward problems situated at the model's capability boundary. Adopting a combinatorial perspective, Li et al. (2025b) propose Knapsack RL, which frames budget assignment as a classical knapsack problem; by modeling tasks as items with specific costs and values, defined by non-zero gradient probabilities and information gain, this method optimally distributes resources to maximize total learning potential. Complementing these approaches, Sun et al. (2025) introduce DOTS, which predicts adaptive difficulty using a reference set to prioritize samples with pass rates near 0.5, while incorporating rollout replay to curtail generation overhead. To execute difficulty estimation dynamically, Yang et al. (2025b) implement a two-stage mechanism that assesses difficulty via pre-rollouts before rebalancing the budget toward harder queries using a hardness-weighted schedule. Finally, Zheng et al. (2025b) extends this adaptability to the temporal

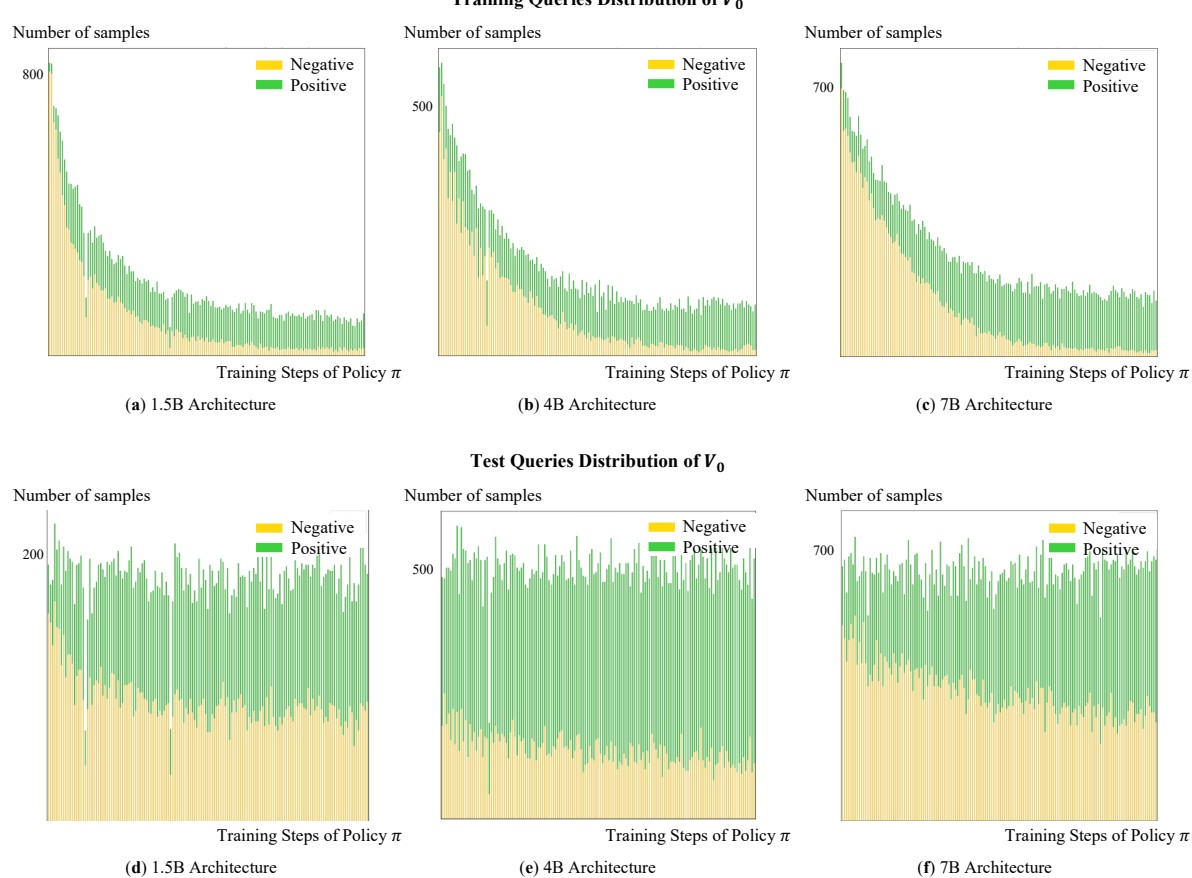

*Figure 7.* **Distribution of Training and Test Queries for $V_0$ across Policy Training Steps.** We visualize the number of positive (green) and negative (yellow) samples utilized for training (Top Row) and testing (Bottom Row) $V_0$ at each checkpoint of the policy $\pi$. The columns correspond to the three distinct architectures: **(a, d)** DeepSeek-R1-Distill-Qwen-1.5B, **(b, e)** Qwen3-4B-Instruct, and **(c, f)** Qwen2.5-7B-Instruct. The x-axis represents the training progress of the policy model. Note that as the policy capability evolves during GRPO training, the ratio of positive to negative samples dynamically shifts, which $V_0$ should track.

dimension by leveraging historical reward dynamics to filter out prompts with persistent zero-variance and adaptively adjust batch sizes to ensure a sufficient quota of effective gradients.

**LLM Inference Routing via Capability Prediction.**    To circumvent the latency costs of online probing, recent research in inference routing has focused on around offline capability prediction, which estimates model proficiency using static representations. This domain generally splits into *learned latent representations* and *reference-based profiling*. Adopting a collaborative filtering perspective, Zhuang et al. (2025) propose EmbedLLM, which treats the instruction-model relationship as a matrix completion problem, deriving compact embeddings from historical logs to predict performance gaps. Focusing on transferable features, Chen et al. (2024b) introduce RouterDC, which optimizes a query-aware router by learning distinct model identifiers through gradient backpropagation, effectively encoding model traits into the routing layer. Similarly, Ong et al. (2024) develop Zooter, which augments these learned proxies by training on large-scale human preference data, aligning routing decisions with nuanced quality judgments rather than simple correctness. Complementing these latent approaches, other works explicitly map capability boundaries using *anchor* data. Zhang et al. (2025c) formalize this by representing models via their accuracy distributions across specific benchmark dimensions (*e.g.*, MMLU). To capture more granular semantic dependencies, Zhang et al. (2025b) introduce The Avengers, which partitions the semantic space of a validation set into clusters; routing is then determined by historical F1 scores within the cluster most similar to the incoming query. Finally, addressing the challenge of generalization to new models, Jitkrittum et al. (2025) propose Universal Model Routing (UMR), which utilizes a correctness vector, a binary signature of a model's success on a fixed anchor set, as a universal feature representation to predict compatibility with unseen instructions.

