# OpenReview forum: "$V_0$: A Generalist Value Model for Any Policy at State Zero"
_ICML.cc/2026/Conference — ICML 2026 regular_

### Official Review · Reviewer_rvRX · 2026-03-12

**Soundness:** 2
**Presentation:** 3
**Significance:** 2
**Originality:** 3
**Overall Recommendation:** 5
**Confidence:** 3

**Summary:**

This paper proposes $V_0$, a generalist value model that predicts the expected performance of any policy at the initial state by conditioning on a context of historical query–performance pairs. Instead of learning a policy-specific value function, the model learns a conditional predictor $V(x, C_\pi)$, where $C_\pi$ represents past observations of the policy’s behavior. This formulation aims to avoid the instability of standard critic learning when the policy changes frequently, which is common in RL for large language models.
The model consists of a frozen embedding backbone, a residual adapter to produce structured features, and a TabPFN head that performs in-context probabilistic inference over the context set. The authors argue that naive cross-entropy training leads to shortcut learning that depends too strongly on the context prior, and propose a ranking + soft cross-entropy objective to encourage prompt-level discrimination within the same context.
Experiments are conducted on GRPO training trajectories across several model families. The paper reports improved value prediction accuracy, better tracking of evolving policies, stronger generalization to unseen queries, and useful downstream performance for rollout budgeting and inference-time routing.

**Compliance With Llm Reviewing Policy:**

Affirmed.

**Key Questions For Authors:**

(1) How much of the improvement comes from the in-context formulation itself versus the TabPFN head? It would help to see stronger ablations where the same adapter is paired with simpler prediction heads, or where TabPFN is replaced with other meta-learning or nonparametric predictors.

(2) How robust is $V_0$ when the context $C_\pi$ is smaller, noisier, or stale? The experiments use relatively large and clean contexts constructed from rollout pools. It would be useful to show performance as context size decreases or label quality degrades.

(3) Can the authors clarify the fairness of the critic baselines? The vanilla value model uses a simple linear head with incremental fine-tuning. Were stronger critic architectures or better tracking strategies attempted? Since the paper’s main claim is superiority over coupled value models, stronger baselines would make the comparison more convincing.

(4) In Eq. (6) and Theorem 4.2, the ranking loss is invariant to additive context bias through score differences. Could the authors clarify under what function class this removes the shortcut term? In particular, does the argument only eliminate additive context offsets, or could multiplicative or interaction-based shortcuts still remain in $s(x, C)$?

**Limitations:**

The method focuses on prompt-level value prediction and does not address token-level or process-level credit assignment, which limits its applicability as a general RL value model.
It requires maintaining historical context datasets with rollout-based labels, which introduces storage, compute, and data-quality overhead that is not fully analyzed.
The experiments are mostly within reasoning-style benchmarks and open-model training trajectories, so the generality of the approach to broader settings remains uncertain.
Because predictions depend on past query-performance data, the method may inherit biases from how contexts are constructed or filtered, which could affect routing or budgeting decisions in deployment.

**Strengths And Weaknesses:**

Strengths:

(1) The paper presents a clear and novel formulation of value estimation as context-conditioned capability prediction, which is well motivated by the critic-policy coupling problem in RL for LLMs.

(2) The mutual-information shortcut analysis provides a useful perspective on why naive value training may fail, and the proposed ranking objective is well justified.

(3) The architecture design is coherent, and the adapter + TabPFN combination is supported by ablations.

(4) The empirical section is broad, covering sequential tracking, strict generalization, ablations, and downstream applications, which strengthens the overall contribution.

Weaknesses:

(1) The empirical comparisons bundle multiple design changes, making it difficult to isolate whether the gains come from the in-context formulation, the TabPFN head, or differences in supervision and training setup.

(2) The vanilla value model baseline appears relatively weak, and stronger critic-training alternatives are not extensively explored.

(3) The method relies on curated historical contexts with rollout-based labels, which may limit practicality in settings where such data is sparse or noisy.

(4) The theoretical analysis is suggestive but does not fully rule out other forms of shortcut learning beyond additive context bias.

(5) The approach only predicts prompt-level values at state zero and does not address finer-grained credit assignment.

---

> ### Author Rebuttal · Authors · 2026-03-31
>
> We thank Reviewer rvRX for the constructive feedback.
>
> ---
>
> ## Q1: Ablation of In-context Formulation vs. TabPFN vs. Setup
>
> **In-context Formulation:**
>
> By ablating the context length ($N$) and permuting the context order, we demonstrate that the performance gains indeed validate our context formulation.
>
> | Context $N$ / Permute | AUC (4B) | B. Alloc (4B) | Routing (Acc, Cost) |
> | :--- | :--- | :--- | :--- |
> | N=32 | .538 | - | (.53, 14k) |
> | N=64 | .553 | - | (.56, 16k) |
> | N=128 | .589 | - | (.63, 22k) |
> | N=256 (Default) | .705 | .573 | (.66, 19k) |
> | N=512 | .733 | - | (.67, 18k) |
> | Permuted ($\times3$)| .736±.012 | .570±.038 | (.65±.02, 19k±.6k) |
>
> *Note: Permuted AUC for 1.5B is .915±.017, and for 7B is .887±.017.*
>
> **Inference Head:**
>
> We replaced TabPFN with simpler non-parametric methods and parameterized prediction heads:
>
> | Head Type | AUC-1.5B | AUC-4B | AUC-7B | B.Alloc | Routing | Overhead |
> | :--- | :--- | :--- | :--- | :--- | :--- | :--- |
> | Vanilla VM | .87 | .90 | .83 | .532 $\to$ .51 | - | Huge (Online) |
> | kNN | .82 | .86 | .75 | .549 | (.59, 17k)| Near Zero |
> | Kernel Reg | .85 | .83 | .76 | - | (.58, 17k)| Near Zero |
> | Mean-Pool+DT | .80 | .81 | .72 | - | (.54, 14k)| Low |
> | Mean-Pool+MLP| .80 | .84 | .79 | .552 | (.62, 28k)| Low |
> | Cross-Attn | .86 | .91 | .81 | - | (.63, 25k)| Low |
> | Ours | .91 | .90 | .88 | .573 | (.66, 19k)| Low |
>
> **Supervision Signal:**
>
> Using capability as a context input introduces the shortcut learning problem.
>
> * Table 5 shows pairwise ranking loss alone overfits, while Soft CE alone lacks accuracy. Our composite loss effectively balances discriminability and calibration.
> * Section 4.3 proves standard supervision degrades into a context-prior shortcut. Figure 6 confirms our composite loss drives residual convergence, ensuring the model learns true causal reasoning.
> * Table 6 indicates jointly tuning the TabPFN Head (instead of freezing) drops AUC sharply to .594.
>
> ---
>
> ## Q2: Comparison with Stronger Critic Baselines
>
> To address concerns regarding weak baselines, we supplement our evaluation with stronger critic networks (e.g., deeper MLP, LoRA, adding a Replay Buffer), as well as recent methods like DVPO and VAPO.
>
> | Baseline | AUC-1.5B | AUC-4B | AUC-7B | B.Alloc(Olymp) | AIME 24 | AMC 23 | MATH 500 | Online Cost |
> | :--- | :--- | :--- | :--- | :--- | :--- | :--- | :--- | :--- |
> | Vanilla (1-L Linear)| .87 | .90 | .83 | .532 $\to$ .51 | .407 | .847 | .891 | Huge |
> | VM (3-L MLP) | .89 | .88 | .83 | .541 | .394 | .836 | .895 | Huge |
> | VM (1-L LoRA) | .85 | .87 | .79 | .550 | .408 | .870 | .896 | Large |
> | VM (8-step Replay) | .88 | .89 | .85 | .552 | .425 | .844 | .903 | Large |
> | DVPO | .76 | .82 | .78 | .556 | .435 | .842 | .905 | Small |
> | VAPO (state 0) | - | - | - | .577 | .473 | .872 | .907 | Huge |
> | Ours | .91 | .90 | .88 | .573 | .481 | .897 | .915 | Small |
>
> *(Note: DVPO is unsuitable for Routing as it requires prior inference for each model, incurring prohibitive overhead.)*
>
> ---
>
> ## Q3: Robustness to Sparse, Noisy, or Stale Contexts
>
> | Condition | AUC-1.5B | AUC-4B | AUC-7B | B.Alloc | Routing |
> | :--- | :--- | :--- | :--- | :--- | :--- |
> | 2% Noise | .90 | .90 | .86 | - | (.66, 19k) |
> | 5% Noise | .88 | .86 | .85 | .571 | (.65, 18k) |
> | 10% Noise | .84 | .81 | .82 | - | (.63, 20k) |
> | kNN Sampling | .93 | .93 | .87 | .567 | (.68, 18k) |
>
> Staleness Test (4B Model):
>
> | Staleness | Model | AUC (1st Ep) | AUC (All Eps)| B.Alloc | Routing |
> | :--- | :--- | :--- | :--- | :--- | :--- |
> | Step = 4 | Vanilla VM | .697 | .838 | .528 | - |
> | | Ours | .882 | .909 | .565 | (.64, 28k) |
> | Step = 32 | Vanilla VM | - | .761 | .536 $\to$ .51| - |
> | *(~1 epoch)*| Ours | - | .842 | .559 | (.62, 20k) |
>
> ---
>
> ## Q4: Theoretical Limits of Non-Additive Shortcuts
>
> This is an insightful observation. We acknowledge that Theorem 4.2 strictly assumes an **additively separable function class** (i.e., additive bias). Mathematically, difference ranking loss cannot guarantee eliminating multiplicative or highly non-linear shortcuts.
>
> However, we mitigate this in practice:
>
> 1. Soft CE: Multiplicative shortcuts evade predictions via extreme scaling factors. Jointly applying Soft CE enforces an absolute probability scale, acting as a strong regularizer against such extreme scaling.
> 2. In LLM value modeling, the dominant shortcut is mean-shift (additive bias). If multiplicative shortcuts dominated, within-context prediction variance would fluctuate quadratically with context capability. Empirically, our prediction standard deviations (Std) remain stable across different capability tiers (Contexts), confirming additive shift is the core shortcut we successfully eliminated:
>
> | Capability in Context | Avg Prediction Std |
> | :--- | :--- |
> | Low 20% | .36 |
> | Middle 40% | .39 |
> | Top 40% | .53 |
>
> ---
>
> ## Q5: Generality to Broader Settings
>
> Please refer to our response to Reviewer y8Fm for additional experiments on general reasoning, coding, and agentic tasks.
>
> **Thanks again!**

---

> > ### Author Rebuttal · Reviewer_rvRX · 2026-04-04
> >
> > Thank you for your detailed response. I don't have further questions.

---

> > > ### Author Response · Authors · 2026-04-04
> > >
> > > We sincerely thank you for the insightful feedback. We remain committed to further refining and strengthening the $V_0$. Thanks again!

---

### Official Review · Reviewer_y8Fm · 2026-03-13

**Soundness:** 3
**Presentation:** 3
**Significance:** 3
**Originality:** 3
**Overall Recommendation:** 4
**Confidence:** 3

**Summary:**

This paper proposes **V0**, a value model for LLM reinforcement learning that decouples value estimation from specific policy parameters. Instead of training a critic tied to an evolving policy, V0 predicts success probability conditioned on a query and a set of historical query-performance pairs representing policy capability. The method combines a frozen embedding backbone, a lightweight adapter, and a TabPFN head, and is trained with a composite objective to reduce shortcut learning. Experiments show that V0 better tracks evolving policies than coupled value baselines, improves a budget-allocation-based training application, and is also useful for inference routing.

**Compliance With Llm Reviewing Policy:**

Affirmed.

**Final Justification:**

The additional experiments and clarifications adequately address my main concerns, especially regarding stronger baselines, label quality, and broader empirical support.

I encourage the authors to calibrate some of the broader claims carefully in the final version, particularly around generality and scaling.

Overall, my concerns have been adequately addressed.

**Key Questions For Authors:**

## Key Questions For Authors

1. How does V0 perform when the policy zoo includes policies trained on non-math domains, such as code or general instruction following, or with optimization algorithms beyond GRPO?

2. How sensitive are the main conclusions to the quality of the target value estimates? In particular, do the results remain consistent when avg@10 is replaced with avg@20 or avg@50?

3. Can the authors compare against a stronger coupled value baseline with parameter-efficient tuning, so that the benefit of decoupling is more cleanly isolated from the benefit of a lighter tuning regime?

4. Do the end-task gains from V0-guided training persist for substantially larger LLMs, and are there any useful scaling trends with respect to the value model size itself?

**Limitations:**

The paper discusses some limitations, but the discussion could be more complete. In particular, the authors should more clearly acknowledge the limited evidence on larger-model training gains and scaling, as well as the risk that imperfect capability estimates could bias training-time allocation or routing decisions.

**Strengths And Weaknesses:**

**Strengths:**

1. **The paper tackles an important and real problem in LLM RL.**
   The motivation is strong: conventional value models are tightly coupled to policy parameters, which makes them costly and brittle under non-stationary policy updates. Reframing the problem as capability-conditioned prediction is a meaningful and well-motivated shift.

2. **The central idea is conceptually novel.**
   The most important contribution is not just a better estimator, but a new formulation: representing a policy through historical behavior and treating value estimation as conditional inference rather than policy-specific function fitting. This is a clean and interesting perspective.

3. **The method design is coherent and aligned with the motivation.**
   The paper does a good job connecting its shortcut-learning analysis to the final training objective. The ranking loss is not added heuristically; it is introduced to reduce context-only bias, while cross-entropy preserves calibration.

4. **The empirical evaluation is relevant to the main claim and goes beyond pure estimation metrics.**
   The experiments focus on the right question—whether the model can track evolving policies—rather than only reporting static held-out accuracy. In addition, the paper demonstrates practical utility in budget allocation during training and inference routing at deployment time.

**Weaknesses:**

1. **The “generalist / any policy” claim is stronger than the current evidence.**
   While the framing is ambitious and appealing, the main training and evaluation are still concentrated in a relatively narrow RLVR setting, mainly involving GRPO-trained policies on math reasoning tasks with verifiable rewards. The results convincingly demonstrate generalization across evolving checkpoints and nearby policies in this regime, but they do not yet fully establish that V0 works as a generalist value model for arbitrary policies, domains, or optimization pipelines. I therefore encourage the authors to better calibrate the scope of this claim.

2. **The main results rely on value labels estimated from avg@10, but uncertainty is not sufficiently quantified.**
   Since each query-policy value is estimated from only 10 stochastic rollouts, the supervision can be noisy, especially for examples with intermediate success probabilities. This matters because the paper’s central claim is about reliable capability estimation under policy evolution. However, the paper does not provide enough analysis on label uncertainty, such as error bars, seed variance, or sensitivity to stronger estimators like avg@20/avg@50.

3. **The current baseline comparison does not fully isolate whether the gain comes from decoupling itself or from a more favorable tuning regime.**
   The vanilla coupled value model is trained via incremental full-parameter fine-tuning, while V0 uses a frozen embedding backbone with a lightweight adapter. As a result, part of the observed gain may come from improved parameter efficiency and regularization, rather than purely from the capability-conditioned formulation. A stronger coupled baseline would make the central empirical claim more convincing.

4. **The paper provides only limited evidence on end-task training gains and scaling to larger LLMs.**
   Beyond the core value-estimation experiments, the paper demonstrates downstream training utility mainly through V0-guided budget allocation, where the final policy improvement is shown on a single 4B model. This is useful evidence, but it does not yet establish whether V0 yields similarly meaningful end-task gains when training substantially larger LLMs. Relatedly, the paper does not study scaling with respect to the value model size itself: the main implementation fixes a frozen 0.6B embedding backbone plus a lightweight adapter, and the reported “scaling” evidence mainly concerns context length rather than parameter scaling. Clarifying these training-time and scaling properties would substantially strengthen the paper.

---

> ### Author Rebuttal · Authors · 2026-03-31
>
> We thank Reviewer y8Fm for the insightful feedback.
>
> ## Q1: The *Generalist / Any Policy* Claim
>
> We sincerely thank the reviewer for pointing this out. We acknowledge that our current wording is too aggressive. In the final version, we will adjust the phrasing to clearly define the boundaries of $V_0$'s current applicability. To further demonstrate $V_0$'s generalization potential beyond pure mathematical reasoning, we have conducted additional experiments on code generation, general QA, and agent environments. As shown below, $V_0$ continues to provide stable performance gains across these diverse domains.
>
> | Method | MBPP | GPQA | ALFWorld | Sokoban |
> | :--- | :--- | :--- | :--- | :--- |
> | GRPO | .642 | .355 | .913 | .802 |
> | Budget Alloc. w/ $V_0$ | .701 | .365 | .946 | .851 |
>
> ---
>
> ## Q2: Uncertainty of Relying on avg@10
>
> To verify $V_0$'s sensitivity to the quality of the estimator, we constructed a golden test set based on 50 stochastic rollouts per query. Our new experiments show that models trained on `avg@10` have almost no statistical difference from those trained on `avg@32` or even `avg@50`. This proves that `avg@10` already provides a sufficiently accurate supervision signal. Furthermore, because $V_0$ relies on a context of 256 samples during inference, this macro-level statistic effectively neutralizes the local noise of individual samples.
>
> | Label Quality | Intra-AUC | Olympiad | AIME 24 | AMC 23 | MATH 500 |
> | :--- | :--- | :--- | :--- | :--- | :--- |
> | avg@5 | .886 | .564 | .473 | .865 | .906 |
> | avg@10 (Ours) | .904 | .573 | .481 | .897 | .915 |
> | avg@16 | .909 | .572 | .487 | .894 | .918 |
> | avg@32 | .908 | - | - | - | - |
>
> ---
>
> ## Q3: Fairness of Baseline Comparisons
>
> To cleanly isolate the gains of our decoupling mechanism from the benefits of lightweight tuning, we have included stronger coupled baselines for comparison (including parameter-efficient methods like LoRA, MLP, Replay, and recent methods like DVPO/VAPO). The results demonstrate that $V_0$ maintains a significant and stable performance advantage, even against these strong baselines. Additionally, we have provided variance statistics across different random seeds, which further validates the robustness of our conclusions.
>
> **Stronger Baselines Comparison:**
> | Baseline | AUC(1.5B) | AUC(4B) | AUC(7B) | Olymp | AIME 24 | AMC 23 | MATH500 | Cost |
> | :--- | :--- | :--- | :--- | :--- | :--- | :--- | :--- | :--- |
> | Vanilla | .87 | .90 | .83 | .510 | .407 | .847 | .891 | Huge |
> | 3-L MLP | .89 | .88 | .83 | .541 | .394 | .836 | .895 | Huge |
> | 1-L LoRA | .85 | .87 | .79 | .550 | .408 | .870 | .896 | Large |
> | Replay | .88 | .89 | .85 | .552 | .425 | .844 | .903 | Large |
> | DVPO | .76 | .82 | .78 | .556 | .435 | .842 | .905 | Small |
> | VAPO | - | - | - | .577 | .473 | .872 | .907 | Huge |
> | Ours | .91 | .90 | .88 | .573 | .481 | .897 | .915 | Small |
>
> **Stability Across Random Seeds (Original vs. Strict Generalization):**
> Table 1: Performance on Original Training Trajectories
> | Arch | 1st Ep. AUC | All Eps. AUC | Pair Acc. | Calib. MSE |
> |:---|:---|:---|:---|:---|
> | 1.5B | .889±.007 | .909±.009 | .940±.012 | .078±.017 |
> | 4B | .887±.012 | .905±.015 | .882±.009 | .099±.010 |
> | 7B | .887±.007 | .886±.015 | .959±.016 | .095±.014 |
>
> Table 2: Strict Generalization Performance
> | Arch | Intra AUC | Pair Acc. | Calib. MSE |
> |:---|:---|:---|:---|
> | 1.5B | .713±.011 | .899±.005 | .144±.023 |
> | 4B | .683±.008 | .807±.007 | .125±.015 |
> | 7B | .696±.005 | .835±.009 | .175±.029 |
>
> ---
>
> ## Q4: Scalability to Larger LLMs and End-Task Gains
>
> * **Training Cost**: Training $V_0$ with the 0.6B backbone takes approximately 26 hours on 16 GPUs. While upgrading to a 4B backbone doubles the time, we can significantly accelerate this by pre-caching the embeddings. Because the Residual Query Adapter extracts structured semantics and the Inference Head utilizes axial attention, the newly added parameter overhead (excluding embeddings) is less than 0.2B, keeping computational costs minimal.
> * **Budget Allocation on Larger LLMs**: We have initiated training on a 30B-scale model (Qwen3-30B-A3B-Instruct) on the DAPO-Math-17k dataset with a batch size of 256. Crucially, we did not retrain $V_0$ for this experiment; we directly applied our existing model. This directly demonstrates $V_0$'s strong zero-shot generalization capability to much larger architectures. Preliminary results up to step 55 show that budget allocation guided by $V_0$ continues to bring consistent end-to-end performance improvements to 30B-level LLMs.
>
> | Method (Qwen3-30B, Step 55)| Olympiad | AIME 24 | AMC 23 | MATH 500 |
> | :--- | :--- | :--- | :--- | :--- |
> | GRPO | .558 | .498 | .859 | .920 |
> | Alloc w/ $V_0$ | .576 | .517 | .877 | .926 |
>
> **Thanks again!**

---

> > ### Author Rebuttal · Reviewer_y8Fm · 2026-04-04
> >
> > Thank you for the rebuttal. The additional experiments and clarifications adequately address my main concerns, especially regarding stronger baselines, label quality, and broader empirical support.
> >
> > I encourage the authors to calibrate some of the broader claims carefully in the final version, particularly around generality and scaling.
> >
> > Overall, my concerns have been adequately addressed.

---

> > > ### Author Response · Authors · 2026-04-04
> > >
> > > We will refine our claims regarding generality and scaling in the final version, to be more grounded and factual. We appreciate your valuable comments. Thank you!

---

### Official Review · Reviewer_tymL · 2026-03-22

**Soundness:** 3
**Presentation:** 3
**Significance:** 3
**Originality:** 2
**Overall Recommendation:** 5
**Confidence:** 3

**Summary:**

This paper proposes V0, a context-conditioned value model that represents a policy through historical query-performance pairs and predicts prompt-level success at state zero using an embedding backbone, a Residual Query Adapter, and a TabPFN head. The experiments on three GRPO-trained model families show strong results on policy tracking, held-out generalization, and two downstream applications: dynamic budget allocation and inference routing.

**Compliance With Llm Reviewing Policy:**

Affirmed.

**Final Justification:**

My main concerns have been addressed by the rebuttal. So I raise my rating to accept.

**Key Questions For Authors:**

1. In the strict generalization experiment, why are kNN-contextual and step-wise retrain not included in Table 2 as well?

2. Can the authors clarify whether Tables 1–2 and the routing/budget-allocation results come from single runs or multiple seeds, and report performance variability if available?

**Limitations:**

Overall, I find the paper interesting and practically useful, but I am less convinced by the novelty claim because the paper is conceptually close to prior decoupled value work such as DVPO, yet does not include a direct empirical comparison to it.

**Strengths And Weaknesses:**

Strengths

1. A clear formulation of a policy-conditioned generalist value model, recasting value estimation as context-conditional prediction over historical query-performance pairs is a neat idea.
2. Empirical results seem to validate the main motivation of the proposed algorithm. Table 1 and Figure 3 show consistent gains over the vanilla coupled critic in tracking evolving policies, and Table 2 suggests the model is learning something more transferable than simple prompt memorization.
3. Thoughtful ablation studies help justify major design choices, which improves credibility.

Weaknesses

1. Closest prior-art comparisons are missing. In particular, it does not empirically compare against the most relevant decoupled value/capability methods discussed in the related work.
2. Less conceptually novel once compared against DVPO’s GVM.
3. The paper does not report whether the main results are averaged across multiple random seeds, and I could not find seed counts, variance estimates, or confidence intervals.

---

> ### Author Rebuttal · Authors · 2026-03-31
>
> We thank Reviewer tymL for the insightful feedback.
>
> ## Q1: Comparison with Related Methods (e.g., DVPO)
>
> We agree that both DVPO and $V_0$ share the core objective of building a decoupled/global value model that eliminates the need for online updates during the RL training. However, to achieve a truly **Generalist** model, the central challenge lies in **how to encode the capability of the current policy ($\pi$) as an input to the value model**. Without explicitly conditioning on the specific policy $\pi$, value estimation $V^\pi(s_0)$ collapses into a pure Reward Model that merely assesses prompt difficulty. Compared to DVPO, $V_0$ introduces fundamental differences and innovations in **capability representation, architecture, and optimization**:
>
> * **Capability Representation of Policy (Prefix vs. Ours Context History):** DVPO implicitly infers policy capability from the generated sequence prefix $s_t = [x, y^{<t}]$. However, as shown in our reasoning case study (in Appendix E), LLMs with vastly different capabilities (e.g., 1.5B vs. 4B) often generate identical correct prefixes but diverge drastically in subsequent complex reasoning steps. Thus, prefixes alone are insufficient to probe true policy capability. $V_0$, instead, reformulates value estimation as a conditional probability prediction $V(C_\pi, s_0)$. Before any text generation (State Zero), it directly ingests 256 historical prompt-reward pairs ($C_\pi$) from the policy as its capability representation. $V_0$ does not evaluate *how well a response is written*; rather, it performs: predicting whether the policy model's capability boundary covers the new query $s_0$.
> * **Architecture:** DVPO uses a standard Transformer with a scalar prediction head, utilizing TD or MSE to regress reward values. This traditional architecture cannot handle the massive $C_\pi$ input required by $V_0$ (256 prompts of ~1k tokens each would result in a 256k context length, causing extreme overhead). Therefore, $V_0$ proposes a novel architecture **using a Residual Query Adapter to extract and heavily compress structured semantic information**, feeding it into a highly efficient axial-attention inference head (TabPFN).
> * **Optimization Trap (Shortcut Trap):** DVPO does not deeply analyze the estimation bias that occurs when evaluating policies of varying strengths. Through Mutual Information analysis, we reveal a deep optimization trap: a naive model easily degenerates into a shortcut heuristic that judges overall policy strength solely from $C_\pi$ while ignoring the specific query $x$'s difficulty. $V_0$ addresses this by designing an effective debiasing loss function.
>
> **Empirical Comparison:**
> We have added direct comparisons with DVPO and other strong baselines. As shown below, $V_0$ consistently leads across all architectures in AUC, downstream dynamic budget allocation (Olymp), and multiple test sets, while maintaining minimal online inference cost.
>
> | Method | AUC-1.5B | AUC-4B | AUC-7B | B.Alloc | AIME24 | AMC23 | MATH500 | Cost |
> |:---|:---|:---|:---|:---|:---|:---|:---|:---|
> | Vanilla VM | .87 | .90 | .83 | .532$\to$.51 | .407 | .847 | .891 | Huge |
> | VM (LoRA)| .85 | .87 | .79 | .550 | .408 | .870 | .896 | Large |
> | DVPO | .76 | .82 | .78 | .556 | .435 | .842 | .905 | Small |
> | VAPO (state0)| - | - | - | .577 | .473 | .872 | .907 | Huge |
> | Ours ($V_0$) | **.91** | **.90** | **.88** | **.573** | **.481** | **.897** | **.915** | **Small** |
>
> ---
>
> ## Q2: Baselines in Strict Generalization (Table 2)
>
> We have included the experimental results for kNN-Contextual and Step-wise Retrain VM under the Strict Generalization setting.
>
> | Method | 1.5B (Intra/Pair/MSE) | 4B (Intra/Pair/MSE) | 7B (Intra/Pair/MSE) |
> |:---|:---|:---|:---|
> | kNN | .672 / .884 / .206 | .643 / .760 / .372 | .674 / .803 / .277 |
> | Step-wise Retrain VM | .528 / .389 / .280 | .533 / .318 / .368 | .540 / .562 / .395 |
>
> ---
>
> ## Q3: Random Seeds
>
> The results in Tables 1, 2, and the downstream tasks in the original paper were from a single run. To improve the reliability of our conclusions, we repeated the experiments using 3 different random seeds and calculated the confidence intervals.
>
> Table 1: Performance on Original Training Trajectories
> | Arch | 1st Ep. AUC | All Eps. AUC | Pair Acc. | Calib. MSE |
> |:---|:---|:---|:---|:---|
> | 1.5B | .889±.007 | .909±.009 | .940±.012 | .078±.017 |
> | 4B | .887±.012 | .905±.015 | .882±.009 | .099±.010 |
> | 7B | .887±.007 | .886±.015 | .959±.016 | .095±.014 |
>
> Table 2: Strict Generalization Performance
> | Arch | Intra AUC | Pair Acc. | Calib. MSE |
> |:---|:---|:---|:---|
> | 1.5B | .713±.011 | .899±.005 | .144±.023 |
> | 4B | .683±.008 | .807±.007 | .125±.015 |
> | 7B | .696±.005 | .835±.009 | .175±.029 |
>
> **Thanks again!**

---

> > ### Author Rebuttal · Reviewer_tymL · 2026-04-06
> >
> > Thank you for the careful rebuttal and added experiments. My concerns have been addressed. I will raise my rating accordingly.

---

> > > ### Author Response · Authors · 2026-04-06
> > >
> > > Thank you for your invaluable feedback. Your suggestions have truly strengthened our paper.
> > > We promise to fully integrate the new baselines and multi-seed results into the camera-ready. We will keep refining the $V_0$ to present the best final version.
> > > Thanks again for your time and support!

---

### Decision · Program_Chairs · 2026-04-30

**Decision:**

Accept (regular)

**Comment:**

The committee is pleased to recommend the acceptance of your work. Your reframing of value models as in-context capability learners offers an elegant solution to the non-stationarity problems typically found in RLVR. Your shortcut analysis using Mutual Information was particularly well-received as a principled way to guide the model's optimization.
Thank you for engaging so throughly during the rebuttal: the additional experiments on 30B models and non-math benchmarks effectively addressed the concerns around $V_0$ being more than a specialized math-solver; it is a step toward a generalist capability-prediction framework. While we acknowledge that extending this to token-level process supervision remains an important future challenge, the current results for budget allocation and inference routing are highly practical and significant. We encourage you to fully integrate the multi-seed variance and the expanded baseline results into your final camera-ready version to ensure the highest standards of clarity and reproducibility